# A nanobody-based toolset to investigate the role of protein localization and dispersal in *Drosophila*

Stefan Harmansa[1†], Ilaria Alborelli[1†], Dimitri Bieli[1], Emmanuel Caussinus[1,2], Markus Affolter[1*]

[1]Growth and Development, Biozentrum, University of Basel, Basel, Switzerland; [2]Institute of Molecular Life Sciences, University of Zurich, Zurich, Switzerland

**Abstract** The role of protein localization along the apical-basal axis of polarized cells is difficult to investigate in vivo, partially due to lack of suitable tools. Here, we present the GrabFP system, a collection of four nanobody-based GFP-traps that localize to defined positions along the apical-basal axis. We show that the localization preference of the GrabFP traps can impose a novel localization on GFP-tagged target proteins and results in their controlled mislocalization. These new tools were used to mislocalize transmembrane and cytoplasmic GFP fusion proteins in the *Drosophila* wing disc epithelium and to investigate the effect of protein mislocalization. Furthermore, we used the GrabFP system as a tool to study the extracellular dispersal of the Decapentaplegic (Dpp) protein and show that the Dpp gradient forming in the lateral plane of the *Drosophila* wing disc epithelium is essential for patterning of the wing imaginal disc.

*For correspondence: markus. affolter@unibas.ch

†These authors contributed equally to this work

**Competing interests:** The authors declare that no competing interests exist.

## Introduction

Despite of its importance, the role of protein localization and the effects of forced protein mislocalization have not been studied extensively and hence remain in many cases not well understood. Over the last few years, genetically encoded protein binders have been introduced to basic biological research and provide novel means for protein manipulation in vivo. While protein function was largely studied by genetic manipulation at the DNA or RNA levels in the past, protein binders allow direct, specific and acute modification and interference of protein function in vivo (*Kaiser et al., 2014*; *Bieli et al., 2016*) and might therefore represent valid tools to study protein localization.

Several types of protein binders exist (for recent reviews see *Helma et al., 2015*; *Plückthun, 2015*). One class of widely applied protein binders are the so-called nanobodies, which are derived from single chain antibodies found in members of the Camelid family. A nanobody specifically recognizing GFP (vhhGFP4, *Saerens et al., 2005*) has been extensively used for cell and developmental biology applications. Importantly, vhhGFP4 functions in the intracellular environment and can be fused to other proteins without losing its activity and specificity in vivo (*Rothbauer et al., 2008*). As a consequence, vhhGFP4 has been functionalized by fusing it to different protein domains in order to visualize (*Rothbauer et al., 2006*), relocalize (*Berry et al., 2016*) and degrade (*Caussinus et al., 2012*; *Shin et al., 2015*) GFP-tagged proteins of interest. More recently, GFP nanobodies were used to generate inducible tools that allow controlled transcription (*Tang et al., 2013*) and enzyme activity (*Tang et al., 2015*), and to generate synthetic receptors (*Harmansa et al., 2015*; *Morsut et al., 2016*), to mention only a few examples.

Recently, we utilized vhhGFP4 to create a synthetic receptor for GFP-tagged signaling molecules and termed it morphotrap (*Harmansa et al., 2015*). Morphotrap consists of a fusion protein between vhhGFP4 and the mouse CD8 transmembrane protein, designed such that the nanobody is

presented extracellularly along the surface of cells. In combination with a GFP-tagged version of the Decapentaplegic (eGFP-Dpp) morphogen, morphotrap proved to be a powerful tool to modify and study secretion and extracellular dispersal of eGFP-Dpp in the *Drosophila* wing disc tissue (*Harmansa et al., 2015*).

Here, we introduce the GrabFP (grab Green Fluorescent Protein) toolbox, consisting of morphotrap and five novel synthetic GFP-traps that either localize to both the apical and basolateral compartment (morphotrap) or preferentially to one compartment: apical (GrabFP-A) or basolateral (GrabFP-B, *Figure 1A*). For each of these three localizations, two versions were constructed in which the vhhGFP4 domain either faces the extracellular space (GrabFP$_{Ext}$) or the intracellular milieu

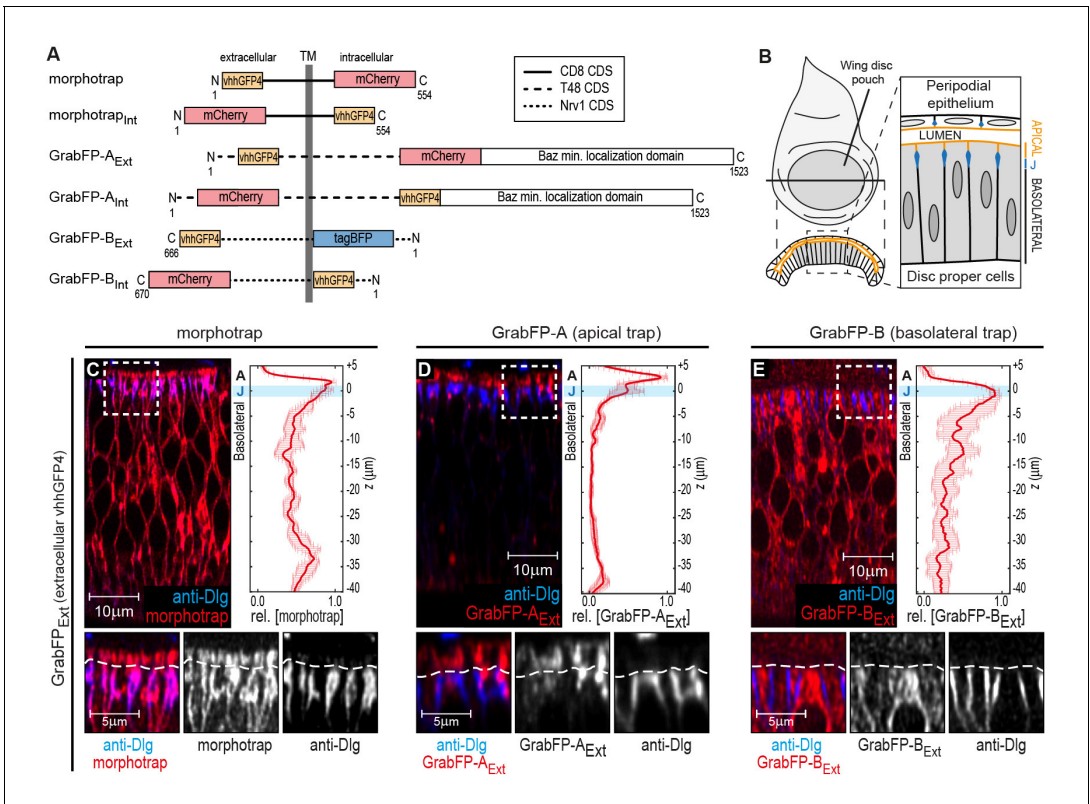

**Figure 1.** The GrabFP constructs localize to distinct regions along the apical-basal axis. (A) Linear representation of the six different versions of the GrabFP system; the constructs exist in two topologies with the GFP-nanobody (vhhGFP4) either facing extracellular (Ext) or intracellular (Int). Numbers refer to the amino acid positions from the N-terminus (N) to the C-terminus (C). TM = transmembrane domain, CDS=coding DNA sequence. (B) Schematic representation of wing disc morphology, the junctions (J) are marked in blue. (C–E) Cross-sections of wing discs expressing morphotrap (C), GrabFP-A$_{Ext}$ (D) and GrabFP-B$_{Ext}$ (E) in the wing pouch (*nub::Gal4*). The GrabFP tools are shown in red and the junctions are visualized by staining for Dlg (blue). In the magnifications the junctional level is marked by a dashed line. Relative distribution of the GrabFP tools along the A-B axis in respect to the junctions (marked by Dlg) is quantified in the plots to the right ($n$ = 4 for each plot, error bars represent the standard deviation). For details on the quantification see Materials and methods and *Figure 1—figure supplement 3*.

The following source data and figure supplements are available for figure 1:

**Figure supplement 1.** Localization of the GrabFP$_{Intra}$ tools.

**Figure supplement 2.** Expression of the GrabFP system allows normal wing development.

**Figure supplement 2—source data 1.** Source data for wing area quantification.

**Figure supplement 3.** Quantification and analysis of protein distribution along the A-B axis.

(GrabFP$_{Int}$). Consequently, the GrabFP system can be used to interfere with target proteins in the extracellular and the intracellular space (*Figure 1A*).

In the following, we first investigate the potential of these anchored GFP-traps to interfere with target protein localization within a cell along the apical-basal (A-B) axis. Our results show that the GrabFP system can effectively mislocalize GFP/YFP-tagged proteins in a controlled manner. As a proof of principle experiment, we characterized the phenotypical consequences of mislocalizing the Myosin II regulatory light chain Spaghetti-squash. In a second part, we use the GrabFP system to trap the Decapentaplegic (Dpp) morphogen in the extracellular space and modify its dispersal in specific regions of the tissue. Our results suggest that the functional Dpp morphogen gradient forms in the lateral plane of the wing disc epithelium.

## Results

### The GrabFP system consists of localized GFP-traps

Analogous to morphotrap, the novel GFP-traps GrabFP-A and GrabFP-B are fusion proteins consisting of vhhGFP4 fused to transmembrane proteins determining the localization and to a fluorescent protein as a marker (*Figure 1A*). All constructs of the GrabFP system were implemented as Gal4 and LexA-inducible transgenes (see Materials and methods).

To test the localization and function of the GrabFP system, we made use of the *Drosophila* wing imaginal disc epithelium, a well-characterized model system to study epithelial polarity (*Tepass, 2012*; *Flores-Benitez and Knust, 2016*) and dispersal of extracellular signaling proteins, for example morphogens (*Thérond, 2012*; *Gradilla and Guerrero, 2013*; *Akiyama and Gibson, 2015*; *Langton et al., 2016*). The wing imaginal disc consists of two contiguous, monolayered epithelial sheets, the pseudo stratified disc proper (DP) epithelium and the squamous peripodial epithelium (PPE; see *Figure 1B*). The apical surface of both, the DP and the PPE, is facing a luminal cavity formed between them. In this study, we characterized the expression and activity of the GrabFP toolset focusing on the columnar cells of the DP epithelium, which will form the adult wing. Visualization of the junctions via the localization of the septate junction component Discs-large (Dlg, see Materials and methods) was used to mark the border separating the apical and basolateral compartment in DP cells.

In order to restrict the GFP-traps to specific regions along the A-B axis, the GFP-nanobody was fused to a protein of known subcellular localization. Morphotrap, based on the mouse CD8 protein scaffold, was shown to localize to both the apical and the basolateral domains (see *Figure 1C* and *Harmansa et al., 2015*). The morphotrap$_{Int}$ construct, in which the nanobody faces the cytosol, also localizes to the apical and basolateral compartments (*Figure 1—figure supplement 1A*).

In order to generate an apically anchored trap (GrabFP-A), we made use of the transcript 48 (T48) protein (*Kölsch et al., 2007*). However, since a fusion protein between the GFP-nanobody, T48, and mCherry showed only mild apical enrichment (not shown), we additionally attached the minimal localization domain of Bazooka (*Krahn et al., 2010*) to the C-terminus of the fusion protein (see *Figure 1A* and Materials and methods for details). Expression in DP cells of both versions of GrabFP-A, GrabFP-A$_{Ext}$ and GrabFP-A$_{Int}$, resulted in strong enrichment in the apical compartment, while only minor amounts of GrabFP-A$_{Ext}$ or GrabFP-A$_{Int}$ were observed along the basolateral domain (*Figure 1D* and *Figure 1—figure supplement 1B*).

Our basolaterally anchored GFP-trap GrabFP-B is based on the Nrv1 protein scaffold (*Figure 1A*, *Sun and Salvaterra, 1995*; *Xu et al., 1999*). Nrv1 localizes to the basolateral compartment of the wing disc, even when overexpressed (*Genova and Fehon, 2003*; *Paul et al., 2007*). In DP cells, GrabFP-B$_{Ext}$ and GrabFP-B$_{Int}$ exclusively localized to the basolateral compartment with no detectable signal along the apical compartment (*Figure 1E* and *Figure 1—figure supplement 1C*).

Expression of the GrabFP constructs in the wing imaginal disc yielded viable and fertile adults with proper wing blade size (*Figure 1—figure supplement 2*), suggesting that the GrabFP system is inert in the absence of GFP and can be used as a tool to study protein function along the A-B axis in the wing imaginal disc.

## Mislocalizing transmembrane and cytosolic proteins along the A-B axis using the GrabFP system

We wanted to test whether the interaction between our localized GFP-traps and a GFP-tagged target protein, transmembrane or cytosolic, can result in defined mislocalization of the target protein. Therefore, single components of the GrabFP system were co-expressed with different target proteins in defined domains of the wing imaginal disc (*hh::Gal4* for GrabFP$_{Ext}$ and *ptc::Gal4* for GrabFP$_{Int}$), while neighboring areas were used as an internal control for the analysis of wild-type target protein localization. We analysed and measured the changes in distribution along the A-B axis of a total of 15 GFP/YFP-tagged proteins, of which 11 were transmembrane/membrane-anchored and four were cytoplasmic proteins. We used target proteins localizing either exclusively to a subcellular compartment (apical or basolateral) or, alternatively, throughout the A-B axis. In order to represent target protein localization, we plotted GFP/YFP fluorescence along the A-B axis.

However, it is known that the binding of nanobodies can interfere with the fluorescent properties of GFP (*Kirchhofer et al., 2010*). We therefore tested if binding of vhhGFP4 to eGFP results in changes of eGFP fluorescence in vitro. Indeed, we observed that binding of vhhGFP4 to eGFP modulated the fluorescent properties of eGFP and resulted in a 1.47-fold increase in eGFP fluorescence in vitro (see *Figure 2—figure supplement 2* and Materials and methods for details). Hence, it is important to consider the possibility that binding of GFP/YFP to our GrabFP traps results in modulation of GFP/YFP fluorescence in vivo. In such a scenario, the observed increase in fluorescence due to GrabFP-mediated mislocalization would be an overestimation of real protein levels. To account for this likeliness in our quantifications, we included A-B profiles of the observed GFP/YFP fluorescence levels (continuous red line) as well as profiles that were corrected for a potential fluorescence increase at GrabFP-positive positions (dashed red line, see Materials and methods for details).

We tested the GrabFP$_{Ext}$ system, which displays the anti-GFP nanobody along the extracellular side (*Figure 2A*), in combination with eight transmembrane proteins extracellularly tagged with GFP/YFP. Expression of either GrabFP-A$_{Ext}$ (*Figure 2B,C* and *Figure 2—figure supplement 1A–B*) or GrabFP-B$_{Ext}$ (*Figure 2D,E,F* and *Figure 2—figure supplement 1C–D*) caused significant changes in the distribution of all eight proteins tested. Generally, GrabFP-A$_{Ext}$ efficiently induced mislocalization of target proteins (i.e. the gain of a novel apical fraction in proteins excluded from the apical compartment, as seen for NrxIV-YFP, *Figure 2B*) and stabilization of an existing apical fraction (as seen for Dlp-YFP, Dally-YFP, PMCA-YFP, *Figure 2C* and *Figure 2—figure supplement 1A–B*). However, GrabFP-A$_{Ext}$ expression did not result in efficient depletion of the basolateral protein fraction (see plots *Figure 2B–C*). This might be due to the fact that GrabFP-A$_{Ext}$ itself was partially mislocalized by the interaction with polarized target proteins and showed enhanced localization to the basolateral compartment (*Figure 2—figure supplement 1E*). In contrast, GrabFP-B$_{Ext}$ displayed a strong potential in depleting apical target-protein fractions (*Figure 2D–F* and *Figure 2—figure supplement 1C–D*). In particular, GrabFP-B$_{Ext}$ significantly reduced the apical pool and increased the basolateral fraction of Dally-YFP, Notch-YFP, Fra-YFP, Crb-GFP and Ed-YFP. Furthermore, GrabFP-B$_{Ext}$ was resistant to mislocalization induced by target protein-interaction (*Figure 2—figure supplement 1F*).

In summary, expression of GrabFP$_{Ext}$ components leads to significant mislocalization of target proteins. Moreover, GrabFP-B$_{Ext}$ caused significant and efficient depletion of the apical fractions of all proteins analyzed.

In a next step, we tested the mislocalization potential of the GrabFP$_{Int}$ system, in which the anti-GFP nanobody localizes intracellularly (*Figure 3A*). To this aim, we used three transmembrane proteins (Fat, Nrv1, Nrv2) containing an intracellular GFP/YFP tag and 3 GFP/YFP-tagged cytoplasmic proteins (Arm, αCat, Hts). We observed significant changes in the distribution of both transmembrane and cytoplasmic target proteins (*Figure 3B–F*). GrabFP-B$_{Int}$ efficiently depleted the apical fraction of Fat-GFP and induced strong enrichment of its basolateral fraction (*Figure 3B*). In contrast, GrabFP-B$_{Int}$ was less efficient in mislocalizing and depleting the apical fraction of the cytoplasmic proteins αCat-GFP and Arm-GFP (*Figure 3C* and *Figure 3—figure supplement 1A*). Concomitantly, GrabFP-B$_{Int}$ showed a higher tendency to be mislocalized when co-expressed with these two cytosolic targets (*Figure 3—figure supplement 1C*). In contrast, GrabFP-A$_{Int}$ efficiently mislocalized target proteins by decreasing their basolateral concentration and increasing their apical fraction. Notably, all proteins tested in combination with GrabFP-A$_{Int}$ had a strong bias toward the

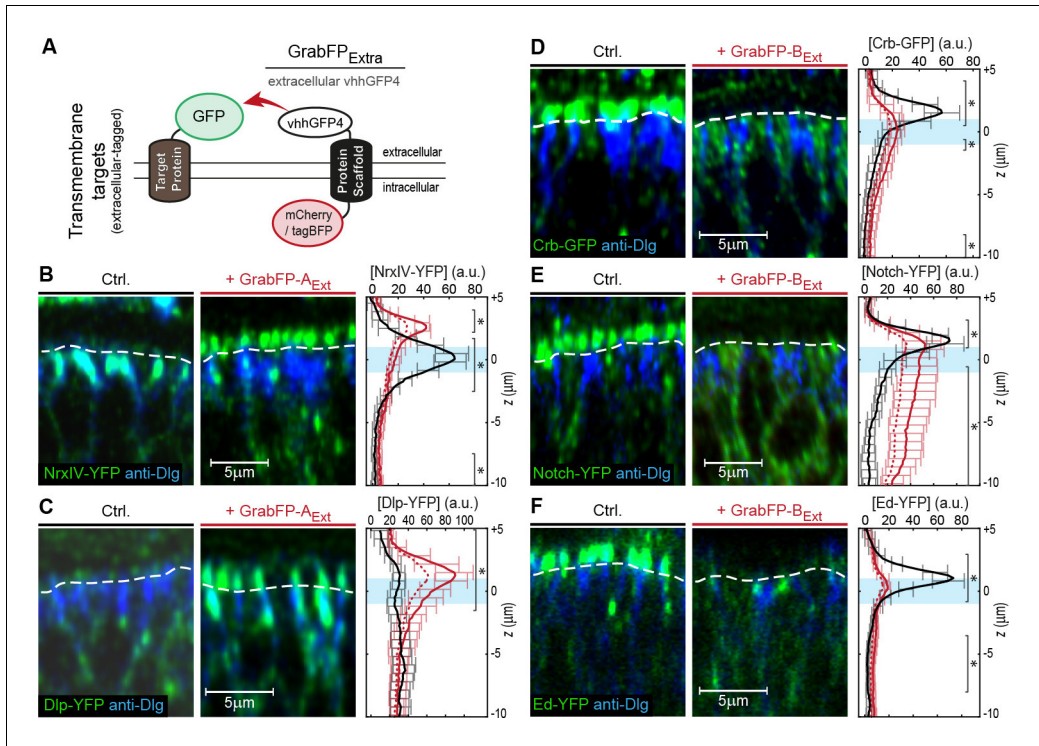

**Figure 2.** Mislocalization of transmembrane proteins using the GrabFP$_{Ext}$ system. (**A**) In the GrabFP$_{Ext}$ system, the GFP-nanobody (vhhGFP4) faces the extracellular space and can interact with extracellular-tagged transmembrane proteins. (**B–C**) Optical cross-section of wing disc cells expressing either NrxIV-YFP (**B**) or Dlp-YFP (**C**) alone (Ctrl., left) or together with GrabFP-A$_{Ext}$ (middle). The junctional level is marked by a dashed line. Quantification of absolute target protein localization (right) along the A-B axis in the absence (black) or in the presence of GrabFP-A$_{Ext}$ (continuous red line). Dashed lines represent profiles corrected for increased GFP/YFP fluorescence due to nanobody binding. The position of the junctions is marked by a blue bar. (Error bars show the standard deviation). (**D–F**) Optical cross sections showing the localization of Crb-GFP (**D**), Notch-YFP (**E**) or Ed-YFP (**F**) in the absence (left) or in the presence of GrabFP-B$_{Ext}$ (middle). Quantifications are shown to the right. (Sample numbers for plots in B-F: NrxIV $n = 10$, Dlp $n \geq 8$, Notch $n \geq 8$, Crb $n = 8$, Ed $n \geq 6$, significance was assessed comparing wild type with corrected profiles using a two-sided Student's $t$-test with unequal variance, *$p<0.05$).

The following source data and figure supplements are available for figure 2:

**Figure supplement 1.** Examples of target protein mislocalization using the GrabFP$_{Extra}$ system.

**Figure supplement 2.** Modulation of EGFP fluorescent properties by vhhGFP4 binding in vitro.

**Figure supplement 2—source data 1.** Source data pannels C-E.

basolateral side in wild-type conditions and acquired a strong apical fraction when co-expressed with GrabFP-A$_{Int}$ (*Figure 3D–F*). Furthermore, GrabFP-A$_{Int}$ showed to be resistant to mislocalization induced by target protein interaction (*Figure 3—figure supplement 1B*).

To further validate the GrabFP system as a tool to study the role of protein localization in vivo, we attempted to mislocalize *Spaghetti squash* (*Sqh*), the *Drosophila* regulatory light chain of Myosin II. We made use of a Sqh-GFP transgene expressed under the control of the *sqh* promoter (*sqhSqh-GFP* flies, *Royou et al., 2004*) that rescues the *sqh^{AX4}* null allele. *Drosophila* Sqh is crucial for morphogenesis and control of epithelial cell shape (*Young et al., 1993*; *Kiehart et al., 2000*). Sqh-GFP is a cytosolic protein that localizes to the subapical cortex in wing disc cells (*Figure 4A*) and is required for maintaining the elongated shape of DP cells (*Widmann and Dahmann, 2009*). To test whether mislocalization of Sqh-GFP from the apical cortex to the basolateral domain affects DP cell shape, we expressed GrabFP-B$_{Int}$ in *sqhSqh-GFP* flies. Expression of GrabFP-B$_{Int}$ in *sqhSqh-GFP*

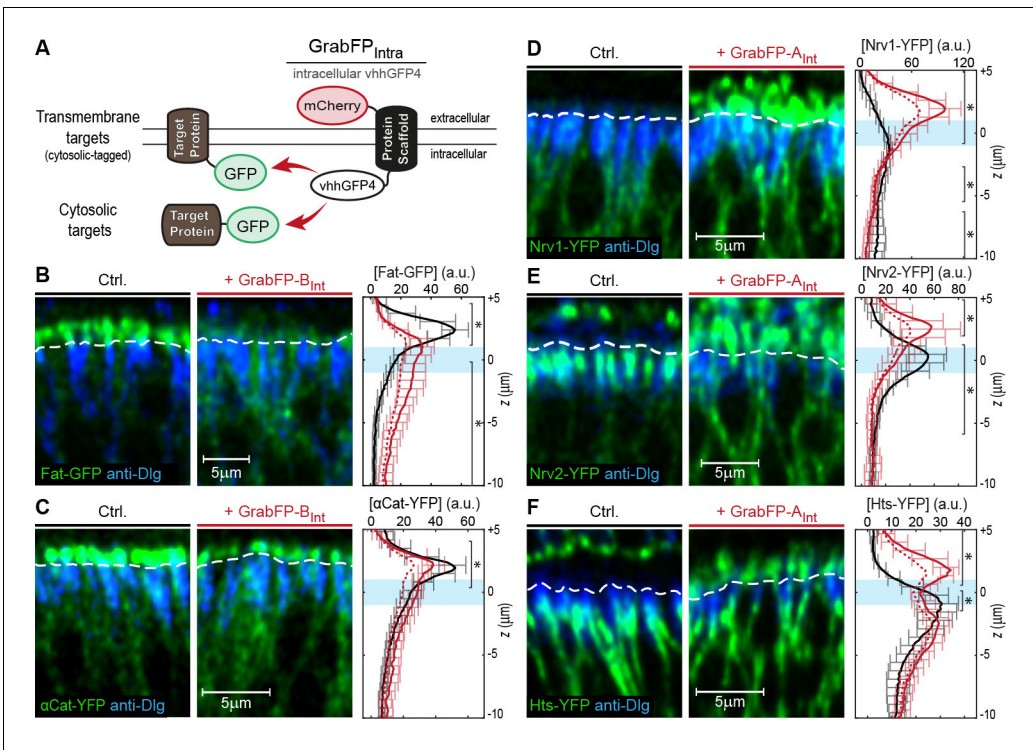

**Figure 3.** Mislocalization of GFP/YFP-tagged proteins using the GrabFP$_{Int}$ system. (**A**) With the GFP-nanobody facing the cytosol, the GrabFP$_{Int}$ system can interact with cytosolic proteins and transmembrane proteins tagged along their cytosolic portion. (**B–C**) Optical cross-sections of wing disc cells expressing either Fat-GFP (**B**) or αCat-YFP (**C**) alone (Ctrl., left) or together with GrabFP-B$_{Int}$ (middle). A dashed line marks the junctional level. Quantification of relative target-protein localization (right) along the A-B axis in the absence (black) or in the presence of GrabFP-B$_{Int}$ (continuous red line). Profiles corrected for fluorescence increase due to nanobody binding are depicted by a dashed red line. The position of the junctions is marked by a blue bar. (Error bars show the standard deviation). (**D–F**) Optical cross-sections showing the localization of Nrv1-YFP (**D**), Nrv2-YFP (**E**) or Hts-YFP (**F**) in the absence (left) or in the presence of GrabFP-A$_{Int}$ (middle). Quantifications are shown to the right. (Sample numbers for plots in **B-F**: Fat $n = 10$, αCat $n = 9$, Nrv1 $n = 10$, Nrv2 $n = 10$, HTS $n = 10$, significance was assessed comparing control with the corrected profiles using a two-sided Student's $t$-test with unequal variance, *$p<0.05$).

The following figure supplement is available for figure 3:

**Figure supplement 1.** Examples of GFP/YFP-target protein mislocalization using the GrabFP$_{Intra}$ system.

female flies that are heterozygous for $sqh^{AX4}$ (and hence, carry one wild-type and one GFP-tagged copy of Sqh) resulted in increased Sqh-GFP levels in the basolateral domain and concomitant reduction in the basal cell surface (***Figure 4B–C***). In $sqhSqh$-GFP male flies, which are hemizygous for $sqh^{AX4}$ (and in which Sqh-GFP represents the only source of Sqh protein), Sqh-GFP mislocalization with GrabFP-B$_{Int}$ caused an even more drastic alteration of cell shape (***Figure 4D***) visible as a strong constriction of the basolateral domain accompanied by a significant expansion of the apical cell surface (***Figure 4F–G***). This behavior could be explained by loss of apical tension (due to the reduction of apical Sqh-GFP) and increased basolateral tension (due to mislocalized Sqh-GFP) (***Figure 4E***). In conclusion, GrabFP-B$_{Int}$ altered the localization of Sqh-GFP, presumably causing significant alterations in the force distribution along the cortex of DP cells.

In summary, our results validate the GrabFP system as novel toolbox to modify protein localization along the A-B axis in a controlled manner and to study the role of protein localization via forced protein mislocalization in vivo.

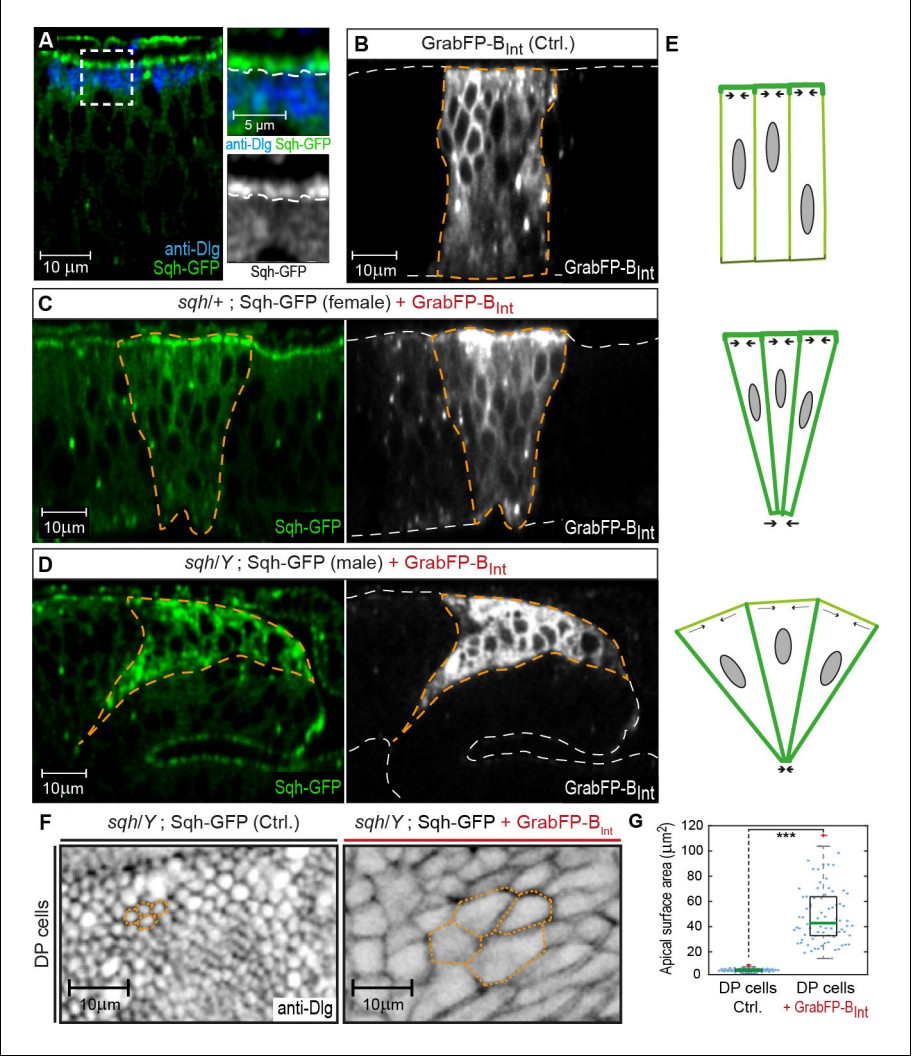

**Figure 4.** GrabFP-B$_{Int}$-mediated Sqh-GFP mislocalization results in changes of DP cell shape. (**A**) Optical cross-section of a wing disc expressing Sqh-GFP (green), stained for Dlg (blue). In the magnifications, the junctional level is marked by a dashed line. (**B–D**) Optical cross-sections of wing discs expressing GrabFP-B$_{Int}$ (grey) in the patched domain (marked by dashed orange line, *ptc::Gal4*) either alone (Ctrl., (**B**) or together with Sqh-GFP (green) in heterozygous *sqh* females (**C**) and hemizygous *sqh* males (**D**). Sqh-GFP mislocalization causes a drastic increase of basolateral Sqh-GFP (**C–D**). Mislocalization of Sqh-GFP causes cell shape alterations resulting in a triangular shape of the *ptc* domain (**C–D**), compared to the rectangular shape of the *ptc* domain in control discs (**B**). The white dashed line marks the apical (top) and basal (bottom) surface of DP cells. (**E**) Schematic representation of the effect of Sqh-GFP mislocalization. Tension is higher in the apical cortex of columnar cells due to polarization of myosin II activity (top). Mislocalization of Sqh-GFP causes increased basolateral tension, leading to constriction of the basolateral cell area (middle). In *sqh* hemizygous conditions, the apical surface expands due to decreased apical myosin II activity (bottom). (**F**) Projections of the junctional level of the DP columnar epithelium of the genotype shown in (**D**) either in the absence of GrabFP-B$_{Int}$ (left, normal Sqh::GFP localization) or in the presence of GrabFP-B$_{Int}$ (right, mislocalized Sqh::GFP). (**G**) Quantification of apical surface area as marked in (**F**). The green line marks the median, statistical significance was assessed using a two-sided Students *t-test* (***p<0.0005), $n \geq 77$.

The following source data is available for figure 4:

**Source data 1.** Source data for apical surface area.

## GrabFP as a tool to study the dispersal of the Decapentaplegic morphogen

Another potential application of the GrabFP system is to study how morphogen gradients form and control patterning and growth during animal development. Morphotrap has previously been used to address the requirement of the Dpp morphogen gradient for patterning and growth of the wing imaginal disc (*Harmansa et al., 2015*). We wanted to extend these studies using the newly generated tools reported here.

A key property that has not been studied in detail is the dispersal of functional Dpp in the wing disc tissue with regard to the A-B axis. We therefore utilize the GrabFP$_{Ext}$ system in combination with an eGFP-tagged version of Dpp (eGFP-Dpp, *Teleman and Cohen, 2000*) to study the localization of the functional Dpp gradient along the A-B axis.

## Dpp disperses in the apical and in the basolateral compartment

In the developing wing imaginal disc, Dpp is expressed and secreted from a central stripe of anterior cells adjacent to the anterior/posterior (A/P) compartment boundary from where it forms a concentration gradient into the surrounding target tissue. The Dpp gradient in the wing disc has been visualized by using different GFP-Dpp fusion proteins (*Entchev et al., 2000*; *Teleman and Cohen, 2000*) and by antibody staining against endogenous Dpp protein (*Gibson et al., 2002*). Dpp was observed in the lateral plane of the wing disc epithelial cells (*Teleman and Cohen, 2000*) as well as apically in the wing disc lumen (*Entchev et al., 2000*; *Gibson et al., 2002*). However, the results of these different studies were not entirely consistent and hence the routes of Dpp dispersal remain controversial.

To investigate the localization of Dpp in the wing disc, we used an eGFP-Dpp fusion protein that was shown to rescue the *dpp* mutant phenotype (*Harmansa et al., 2015*). When eGFP-Dpp is expressed in its endogenous expression domain using the LexA/LOP binary expression system (*dpp::LG*, *Yagi et al., 2010*), it forms a wide concentration gradient into the target tissue (*Figure 5A* and [*Harmansa et al., 2015*]). In order to better characterize the localization of eGFP-Dpp in the wing imaginal disc, we acquired high-resolution confocal stacks along the z-axis. Optical cross-sections revealed that eGFP-Dpp localized prominently to dotted structures along the lateral region of the DP (*Figure 5B*, arrowheads), which were suggested to represent endocytic vesicles (*Teleman and Cohen, 2000*). We did not detect eGFP-Dpp signal within the luminal space (*Figure 5B*, magnification). These results suggest that, using fluorescence microscopy, Dpp is prominently detected within the lateral plane of the DP epithelium.

Morphotrap was reported to immobilize and accumulate eGFP-Dpp on the cell surface (*Harmansa et al., 2015*). Therefore, we used morphotrap to visualize even low levels of extracellular eGFP-Dpp and to determine where along the A-B axis eGFP-Dpp encounters morphotrap-expressing target cells. When we expressed eGFP-Dpp in its central stripe source (using *dpp::LG*) and morphotrap in clones (*Figure 5C*), we observed high amounts of immobilized eGFP-Dpp on the proximal surface (the one facing the source of Dpp) of morphotrap clones situated in the target tissue (*Figure 5D–E*). Subapical projections (*Figure 5D*) as well as optical cross sections (*Figure 5E'*) showed that low amounts of eGFP-Dpp accumulated on the apical surface of morphotrap clones. However, the prominent majority of eGFP-Dpp accumulation was observed along the basolateral cell surface of morphotrap clones (*Figure 5E*). These results suggest that only low amounts of eGFP-Dpp disperse in the apical/luminal compartment while the majority of eGFP-Dpp dispersal takes place along the basolateral compartment.

## GrabFP can specifically interfere with sub-fractions of the Dpp gradient

To investigate the role of apical and basolateral Dpp pools in patterning and growth control, we expressed eGFP-Dpp in the stripe source (using *dpp::LG*) and the different versions of the GrabFP$_{Ext}$ system in the posterior compartment (using *hh::Gal4*, see *Figure 6B–D*, left). Thereby, we specifically interfered with Dpp dispersal in the posterior compartment, not modifying Dpp production and secretion.

As shown before, eGFP-Dpp expressed in a wild-type background is observed in presumptive vesicular structures along the lateral plane of the epithelium but is not present at detectable levels in the wing disc lumen (*Figure 6A*). Posterior morphotrap expression resulted in immobilization of

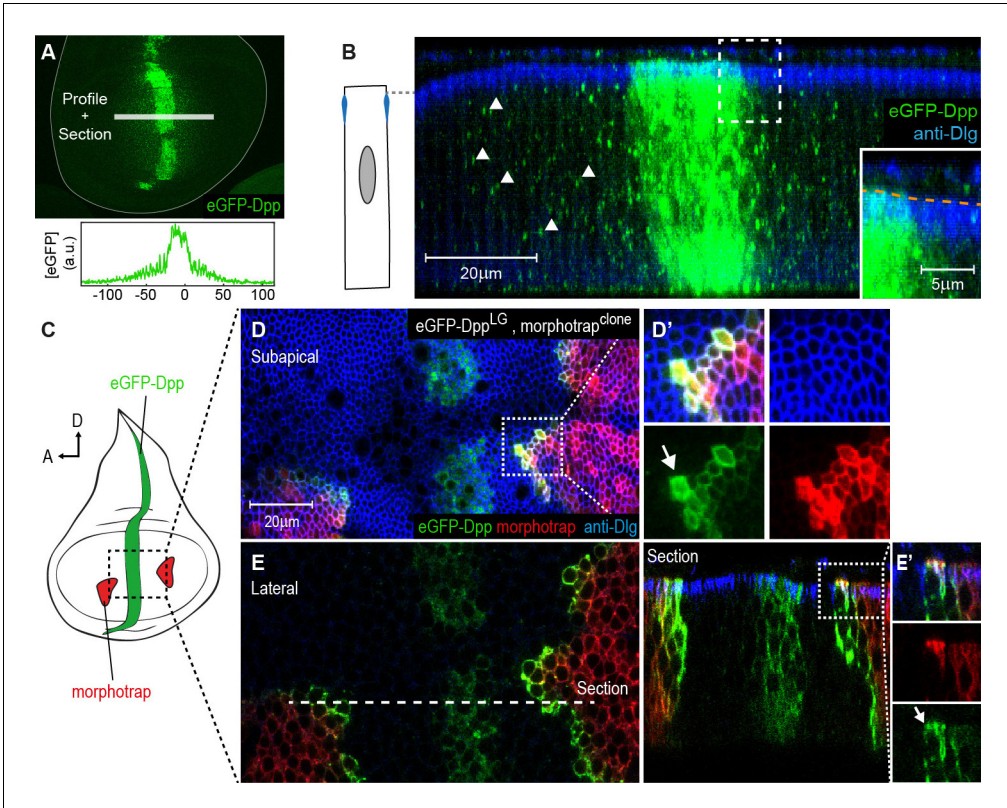

**Figure 5.** The Dpp morphogen spreads in the apical and basolateral compartment. (**A**) Wing disc expressing eGFP-Dpp in the central Dpp stripe and eGFP fluorescence profile (bottom). (**B**) Optical cross-section of a wing disc as shown in (**A**) additionally stained for Dlg (blue). eGFP-Dpp is prominently observed in spots (arrowheads) along the lateral axis of the disc but not in the wing disc lumen (see magnified insert). (**C**) Scheme of morphotrap expression in clones and eGFP-Dpp in the central *dpp* stripe. (**D**) Subapical projection of a wing disc expressing eGFP-Dpp in the *dpp* stripe and two lateral morphotrap clones. Magnifications to the right show apical eGFP-Dpp immobilization on the proximal surface of morphotrap clones. (**E**) Lateral projection of the wing disc shown in (**D**). An optical cross-section to the right shows low level apical (also see arrow in magnification in [**E'**]) and high level basolateral immobilization of eGFP-Dpp.

eGFP-Dpp predominantly along the basolateral compartment of target cells adjacent to the Dpp source. In few cases, eGFP-Dpp immobilization was observed along the apical surface of morphotrap expressing cells (see *Figure 6B*, arrow in right image and *Figure 6—figure supplement 1A*). Since the A/P boundary in the PPE is shifted anteriorly, morphotrap is also expressed in the PPE cells covering the Dpp DP source. Interestingly, PPE cells covering the Dpp DP source showed substantial immobilized eGFP-Dpp on their luminal surface (*Figure 6B*, asterisk in right image). This observation suggests that a fraction of Dpp is secreted into the lumen and disperses in the luminal cavity. These results show that posterior expression of morphotrap reduces spreading of apical and basolateral Dpp pools into the posterior compartment.

Posterior expression of GrabFP-B$_{Ext}$ resulted in the exclusive basolateral immobilization of eGFP-Dpp close to the source (*Figure 6C*), consistent with its restricted localization to the basolateral membrane.

In sharp contrast, posterior expression of GrabFP-A$_{Ext}$ resulted in strong apical and peripodial, but also basolateral immobilization of eGFP-Dpp (*Figure 6D* and *Figure 6—figure supplement 1C*). Therefore, it seems that the relative small portion of GrabFP-A$_{Ext}$ localizing to the basolateral side is large enough to interfere with basolateral eGFP-Dpp dispersal (or that eGFP-Dpp relocalizes GrabFP-A$_{Ext}$). The increased levels of apical eGFP-Dpp immobilization might also hint toward mislocalization of basolateral immobilized eGFP-Dpp to the apical compartment by GrabFP-A$_{Ext}$.

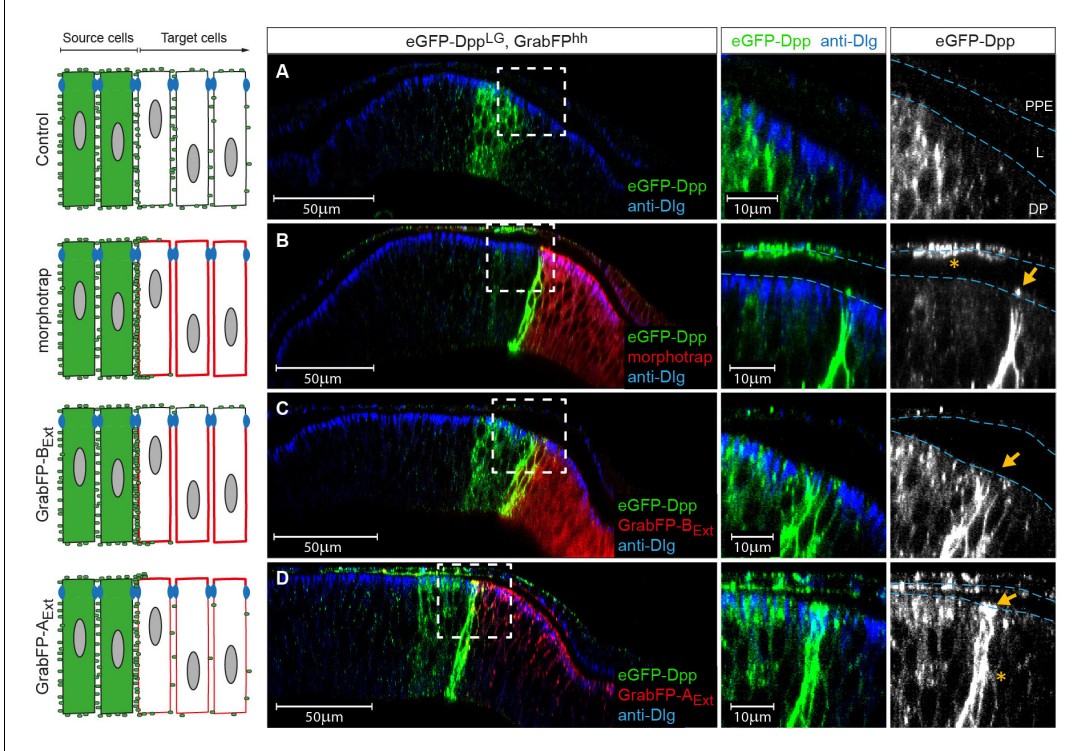

**Figure 6.** The GrabFP$_{Ext}$ system can interfere with specific sub-fractions of the Dpp morphogen gradient. Optical cross sections of wing discs either expressing eGFP-Dpp (green) in the stripe source (**A**) or eGFP-Dpp in the stripe and the different versions of the GrabFP system (red, **B–D**) in the posterior compartment of disc proper and PPE cells (*hh::Gal4*). (**A**) When expressed alone (Ctrl.), eGFP-Dpp is mainly observed in the lateral plane of the DP epithelium. Peripodial epithelium (PPE), lumen (L) and disc proper epithelium (DP). (**B**) Posterior expression of morphotrap results in strong eGFP-Dpp immobilization along the basolateral domain and low or no apical immobilization (see arrow in the magnification to the right). eGFP-Dpp is also immobilized on the apical surface of PPE cells overlaying the Dpp DP source (see asterisk in magnification). (**C**) Posterior expression of GrabFP-B$_{Ext}$ results in exclusive immobilization of eGFP-Dpp in the basolateral domain. No apical immobilization is observed, neither in DP (see arrow) nor in PPE cells. (**D**) Expression of GrabFP-A$_{Ext}$ in the posterior compartment results in strong basolateral (asterisk) and apical (arrow) immobilization of eGFP-Dpp.

The following figure supplement is available for figure 6:

**Figure supplement 1.** Quantification of differential eGFP-Dpp accumulation by morphotrap, GrabFP-B$_{Ext}$ and GrabFP-A$_{Ext}$.

In summary, the GrabFP$_{Ext}$ system can be used to interfere with both apical and basolateral dispersal (morphotrap) or to specifically interfere with basolateral eGFP-Dpp dispersal (GrabFP-B$_{Ext}$). However, localization of GrabFP-A$_{Ext}$ is not specific enough to exclusively interfere with apical Dpp dispersal (see also Discussion).

## Basolateral Dpp dispersal is required for patterning and growth of the *Drosophila* wing

In an earlier study using morphotrap, we reported that Dpp dispersal is important for wing disc growth and patterning (*Harmansa et al., 2015*). Since we find that Dpp is prominently found in the basolateral compartment, we used the newly generated GrabFP system to investigate whether basolateral Dpp dispersal is required for patterning of the wing. We therefore compared the p-Mad signaling response of $dpp^{d8/d12}$ mutant wing discs rescued with eGFP-Dpp (normal Dpp dispersal) to $dpp^{d8/d12}$ mutant wing discs rescued with eGFP-Dpp expressing either morphotrap (apical and basolateral Dpp dispersal reduced) or GrabFP-B$_{Ext}$ (only basolateral Dpp dispersal reduced) in the posterior compartment, respectively (*Figure 7A–H*).

In control conditions (normal Dpp spreading), p-Mad forms a wide bilateral concentration gradient into the anterior and posterior compartments (>40 µm; see *Figure 7A*). In contrast, reduction of

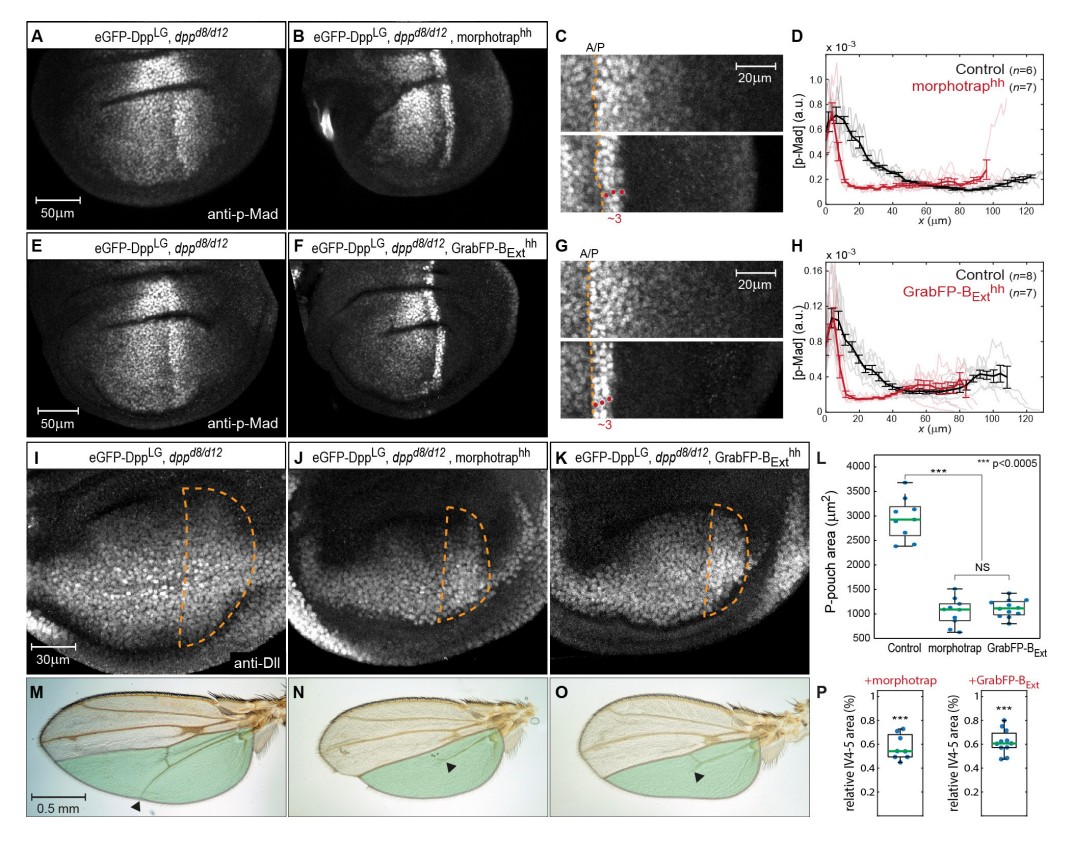

**Figure 7.** Basolateral Dpp spreading is required for patterning and size control. (**A–B**) p-Mad staining in representative $dpp^{d8/d12}$ mutant wing disc rescued by eGFP-Dpp (A) and in $dpp^{d8/d12}$ wing disc rescued by eGFP-Dpp expressing morphotrap in the posterior compartment ($hh::Gal4$, B). (**C**) Magnifications of the posterior, dorsal pouch region of the images shown in (**A–B**). The A/P boundary is marked by a dashed yellow line. (**D**) Average posterior p-Mad profiles of 98–100 hr AEL old $dpp^{d8/d12}$ wing disc rescued by eGFP-Dpp (black) and $dpp^{d8/d12}$ wing disc rescued by eGFP-Dpp expressing morphotrap (red). (**E–H**) Representative wing discs and quantification of p-Mad levels in $dpp^{d8/d12}$ wing disc rescued by eGFP-Dpp (E, black in H) and $dpp^{d8/d12}$ wing disc rescued by eGFP-Dpp expressing GrabFP-B$_{Ext}$ in the posterior compartment (F, red in H). (**I–K**) Representative 98–100 hr AEL old wing discs of the indicated genotypes stained for Distal-less (Dll) as a marked for pouch size. The posterior wing pouch is outlined by a dashed yellow line. (**L**) Quantification of posterior wing pouch area as shown in (**I–K**). (Control $n = 9$, morphotrap $n = 10$, GrabFP-B$_{Ext}$ $n = 12$) (**M–O**), Female wings of the genotypes indicated. The area posterior to vein 4 (IV4-5) is marked in green. Block of apical and basolateral, as well as block of basolateral Dpp dispersal results in a loss of the distal parts of wing vein 5 and hence patterning (see arrowheads). (**P**) Quantification of relative IV4-5 area as indicated in (**M–O**). (***$p > 0.0005$, Control $n = 10$, morphotrap $n = 8$, GrabFP-B$_{Ext}$ $n = 11$).

The following source data and figure supplement are available for figure 7:

**Source data 1.** Wing pouch area and IV4-5 area.

**Figure supplement 1.** morphotrap expression in PPE cells interferes with luminal Dpp spreading.

apical and basolateral spreading by expression of morphotrap in the posterior compartment resulted in a drastic reduction of the posterior p-Mad range to ~3 cells or ~10 µm (**Figure 7B–D**). Interestingly, specifically interfering with basolateral Dpp spreading by posterior expression of GrabFP-B$_{Ext}$ also resulted in a reduction of posterior p-Mad range to ~3 cells or ~10 µm (**Figure 7F–H**), a result strikingly similar to the morphotrap experiment. Hence, these experiments demonstrated that basolateral Dpp spreading is required for proper Dpp signaling range and patterning and that apical/luminal Dpp spreading is not sufficient.

We also investigated whether growth of the wing disc requires basolateral Dpp spreading. Indeed, we found that the posterior wing pouch area visualized by immunostaining against Distal-less (Dll) was reduced to a similar extend when expressing either morphotrap or GrabFP-B$_{Ext}$ in the

posterior compartment (*Figure 7I–L*). Accordingly, the posterior wing blade area was reduced to a similar extend in both the morphotrap and the GrabFP-B$_{Ext}$ conditions (*Figure 7M–P*). In addition, and consistent with the strongly reduced p-Mad range, the distal portion of wing vein 5 was lost upon posterior expression of morphotrap or GrabFP-B$_{Ext}$ (19/19 wings).

In summary, these results show that basolateral, not apical/luminal Dpp dispersal is important for patterning and size control of the wing disc and the adult wing. To further test the requirement of luminal Dpp spreading, we expressed morphotrap in PPE cells to hinder luminal Dpp dispersal (*Figure 7—figure supplement 1*). However, we observed only very mild effects on wing patterning and growth in this condition, supporting the view that apical Dpp spreading plays a minor role in wing development.

## Dpp dispersal in the basal and lateral plane control wing disc growth

Our results suggest a prominent role of basolateral Dpp spreading in the wing imaginal disc. Hence, we further dissected the function of Dpp spreading along the basolateral compartment. The basolateral compartment consists of the lateral region, where epithelial cells are compactly surrounded by their neighbors, and the basal region, where cells contact the extracellular matrix (ECM) of the basal lamina (BL). Dpp is known to interact with the heparin sulphate proteoglycans Dally and Dally-like localizing to the apical and lateral region (*Figure 2C* and *Figure 2—figure supplement 1A*) as well as with Collagen IV localizing to the BL (*Wang et al., 2008*).

In order to investigate the role of Dpp spreading in the lateral plane versus Dpp spreading in the BL, we generated GrabFP-ECM, a GFP-trap localizing to the extracellular matrix of the BL. GrabFP-ECM is a fusion protein consisting of the coding sequence of the *Drosophila* Collagen IV gene *viking* (*vkg*, *Yasothornsrikul et al., 1997*; *Wang et al., 2008*), vhhGFP4 and mCherry inserted between the first and the second exon of *vkg* (see Materials and methods). When expressed in the larval fat body (*r4::Gal4*) GrabFP-ECM integrated into the BL of the wing disc (*Figure 8A–B*) as observed for wild-type collagen IV (*Pastor-Pareja and Xu, 2011*).

When GrabFP-ECM was expressed in the fat body of *dpp$^{d8/d12}$* mutant larvae rescued with eGFP-Dpp (GrabFP-ECM$_{Rescue}$ flies), high levels of eGFP-Dpp were immobilized in the BL underlying the Dpp source and low, graded levels were immobilized in the BL further away from the source stripe (*Figure 8C–D*). Hence, GrabFP-ECM can specifically trap Dpp and affect its dispersal in the BL, while Dpp dispersal in the lateral plane of the disc epithelium remains unaffected (*Figure 8D*).

To study the function of Dpp dispersal in the BL, we compared p-Mad profiles in control discs and in discs of GrabFP-ECM$_{Rescue}$ flies (*Figure 8E–G*). Wing discs of GrabFP-ECM$_{Rescue}$ flies showed a clear reduction in p-Mad range and peak levels (*Figure 8E–G*). The reduction in p-Mad range was accompanied by a significant reduction in wing disc pouch size (*Figure 8H–J*) and adult wing blade area (*Figure 8K–M*). These findings suggest that basally secreted Dpp and/or Dpp spreading in the BL contribute to proper Dpp signaling range and size control. However, despite a clear reduction in size, the overall patterning of the wing seemed unaffected in the GrabFP-ECM condition (*Figure 8L–M*) suggesting that basal Dpp is not strictly required for patterning the fly wing. Yet, quantification of the intervein areas showed that the medial region adjacent to the Dpp source is most susceptible to a reduction of Dpp signaling levels (*Figure 8N*).

## Discussion

Many proteins localize to specific membrane domains or organelles within a cell, and it has been shown in several cases that correct protein localization plays a vital role in cell homeostasis (*Wodarz and Näthke, 2007*; *Mellman and Nelson, 2008*). However, the functional implication and the necessity of proper localization, as well as the consequences of distinct mislocalization of a given protein, are less well understood. Here, we have developed and used a novel, nanobody-based toolset, the GrabFP system, to interfere with the localization of GFP-tagged proteins along the apical-basal axis in the larval wing imaginal disc.

### The GrabFP system can interfere with protein localization

Recently, it was reported that tethering of nanobodies to specific cellular compartments can result in protein relocalization (*Berry et al., 2016*). In line with these observations, expression of the GrabFP constructs altered the subcellular localization along the apical-basal axis of the 15 different GFP-

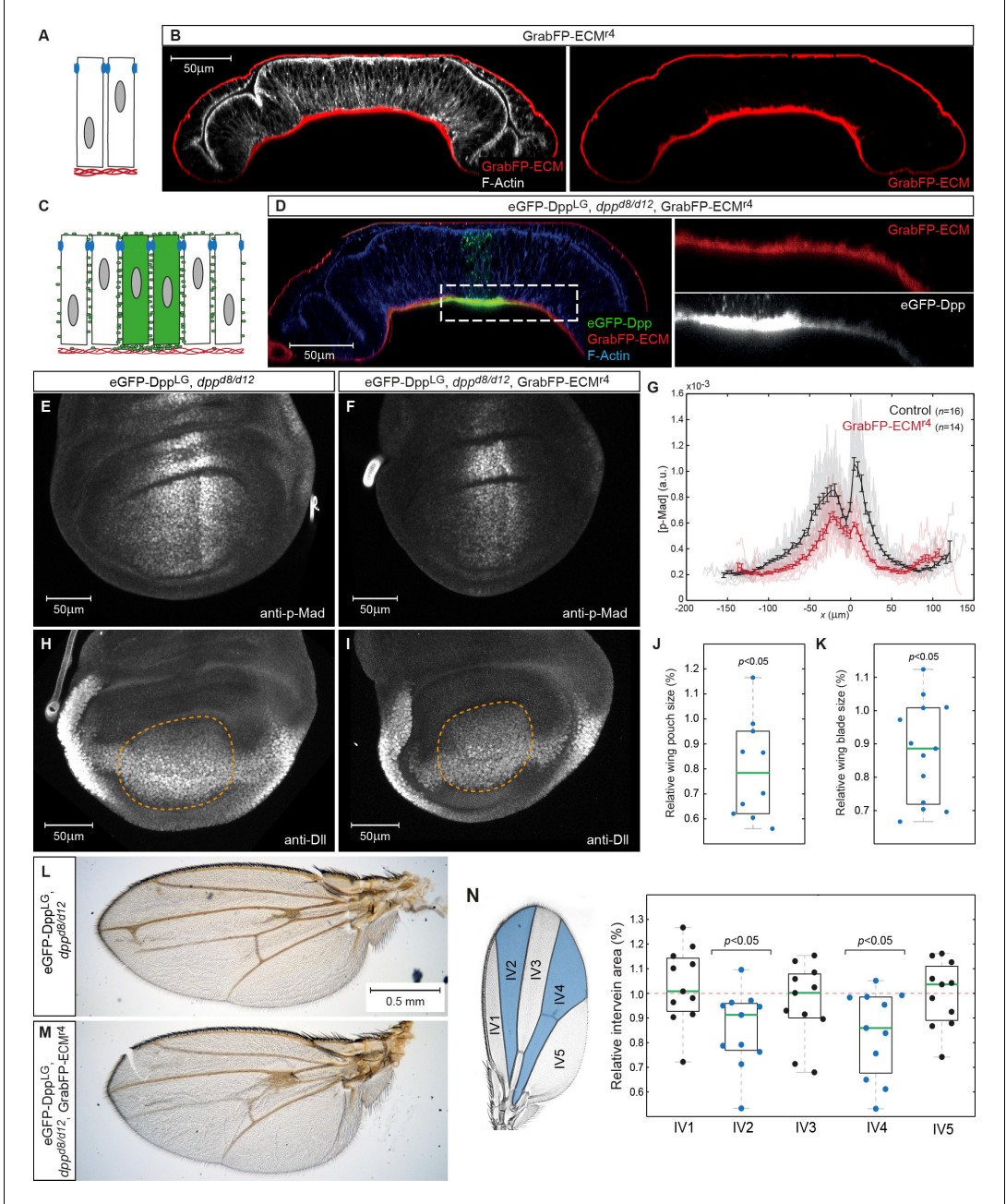

**Figure 8.** Basal Dpp is required to control wing size. (**A**) Schematic representation of GrabFP-ECM localization when expressed in the larval fat body. (**B**) Wing disc optical cross-section of an animal expressing GrabFP-ECM in the fat body, stained for mCherry (GrabFP-ECM, red) and F-Actin (Phalloidin, white). (**C**) Schematic of eGFP-Dpp immobilization in the ECM by GrabFP-ECM. (**D**) Optical cross-section of a *dpp$^{d8/d12}$* mutant wing disc rescued by eGFP-Dpp (green) and GrabFP-ECM (red) localizing to the basal lamina. Tissue outlines are visualized by F-Actin staining (blue). Magnification to the right shows strong eGFP-Dpp accumulation below Dpp source cells. (**E–F**) 98–100 hr AEL old wing discs of the indicated genotype stained for p-Mad. (**G**) Average p-Mad gradient at 98–100 hr AEL. (**H–I**) 98–100 hr AEL old wing discs of the above indicated genotypes stained for Dll. The wing pouch is outlined by a dashed yellow line and quantified in (J). (**J**) Relative wing pouch area of *dpp$^{d8/d12}$* mutant wing disc rescued by eGFP-Dpp and GrabFP-ECM localizing to the basal lamina (*n* = 11). (**K**) Relative wing blade area of *dpp$^{d8/d12}$* mutant wing disc rescued by eGFP-Dpp and GrabFP-ECM localizing to the basal lamina (*n* = 14). (**L–M**) Representative female wings of the genotypes indicated. (**N**) Quantification of intervein area in GrabFP-ECM flies relative to control wings (*n* = 11).

The following source data is available for figure 8:

**Source data 1.** Wing pouch and blade areas.

tagged cytosolic or transmembrane proteins we tested. All the different components of the GrabFP system induced drastic mislocalization of target proteins, causing the gain of a novel subcellular fraction, which was minor or absent in wild-type conditions. In addition, the GrabFP system significantly depleted the wild-type subcellular fractions of two-thirds of the tested target proteins.

An interesting target that was effectively mislocalized is the transmembrane receptor Notch (Notch-YFP). Notch signaling is required for cell-cell communication and differentiation during development (*Guruharsha et al., 2012*). The apical localization of Notch is conserved in different tissues and organisms, suggesting that it is crucial for Notch function (*Fehon et al., 1991*; *Ohata et al., 2011*; *Hatakeyama et al., 2014*). In particular, Notch apical localization might be necessary to allow interaction with its ligand Delta, which also localizes to the apical cell surface (*Sasaki et al., 2007*). In future studies, the GrabFP system will help to better understand the requirements for polarized distribution of signaling pathway components in different developmental contexts.

In line with observations made by *Berry et al.* (*Schornack et al., 2009*; *Berry et al., 2016*), the GrabFP components were in some cases themselves mislocalized due to interaction with target proteins. This was particularly relevant for GrabFP-A$_{Ext}$ and GrabFP-B$_{Int}$ (*Figure 2—figure supplement 1G* and *Figure 3—figure supplement 1H*), which were, presumably as a consequence, less efficient in causing target protein mislocalization.

In conclusion, the GrabFP system provides a general and ready-to-use framework to specifically mislocalize GFP-tagged proteins. Conveniently, large collections of GFP-tagged protein are available in *Drosophila melanogaster* (*Lowe et al., 2014*; *Lye et al., 2014*; *Nagarkar-Jaiswal et al., 2015*; *Sarov et al., 2016*). Moreover, the GrabFP system can be induced in a tissue-specific and temporally controlled manner and thus represents a versatile tool to study the effect of forced protein mislocalization and protein function in specific subcellular compartments in vivo.

## Localized nanobodies to study the functional role of protein localization

Using Sqh-GFP, we have provided a first example of GrabFP implementation for functional studies on protein localization. We have initially described a role of Sqh during dorsal closure in the *Drosophila* embryo using the deGradFP system (*Caussinus et al., 2012*). Tissue-specific degradation of Sqh (which leads to a failure to contract actomyosin networks) combined with laser ablation studies have now shown that amnioserosa cell constriction but not actin cable tension in the adjacent dorsal ectodermal cells autonomously drives dorsal closure (*Pasakarnis et al., 2016*). Similarly, the role of Sqh localization and the effect of Sqh mislocalization on epithelial cell shape can now be studied in more detail using the GrabFP toolset combined with other approaches such as laser ablation and force measurement.

## A basolateral Dpp pool is essential for patterning the *Drosophila* wing imaginal disc

We have previously used morphotrap to show that spreading of eGFP-Dpp is required for wing pouch patterning and for medial growth, while it is dispensable for lateral wing disc growth (*Harmansa et al., 2015*). Based on this finding, we have used the GrabFP system to further dissect the functional role of eGFP-Dpp spreading with regard to the apical-basal axis in *Drosophila* wing disc development. We find that the vast majority of the eGFP-Dpp pool can be immobilized on the basolateral side of disc cells, indicating that Dpp spreads in the basolateral intercellular space. In line with this, functional interference with Dpp spreading in the basolateral compartment only (GrabFP-B$_{Ext}$) suggests that the patterning function of the Dpp gradient is brought about to a large extend by Dpp spreading in the lateral plane of the wing disc epithelium. Growth control, in contrast, is influenced by Dpp dispersing in both the lateral and in the basal planes. These results are based on the findings that restricting basolateral Dpp dispersal (using GrabFP-B$_{Ext}$) strongly impairs pattern and size while immobilizing eGFP-Dpp in the BL (using GrabFP-ECM) only impairs the size of the *Drosophila* wing.

Our finding of a prominent role of the basolateral compartment in Dpp spreading is interesting with regard to the mechanism of gradient formation and, at the same time, raises several new questions. Dpp gradient formation in the *Drosophila* wing disc remains a paradigm to study morphogen dispersal and several mechanisms for morphogen gradient formation have been suggested, operating in different extracellular environments (for a recent review see [*Akiyama and Gibson, 2015*]).

These proposed mechanisms include free extracellular diffusion in the wing disc lumen (*Zhou et al., 2012*), restricted extracellular diffusion in the lateral plane of the epithelium (*Belenkaya et al., 2004*; *Akiyama et al., 2008*; *Schwank et al., 2011*), and active transport by actin-based filopodial extensions called cytonemes along the apical surface of DP cells (*Hsiung et al., 2005*). While the formation of the functional Dpp gradient in the lateral compartment is compatible with a restricted extracellular diffusion mechanism, it is not as easily compatible with the formation of a functional Dpp gradient via free diffusion in the lumen or with a key role of apical cytonemes in Dpp readout. Since we have not been able to visualize apical cytonemes, neither in wild type discs nor in disc, in which Dpp spreading along the basolateral side was blocked, we cannot make firm statements about a direct involvement of apical cytonemes in either situation.

In line with Dpp gradient formation via restricted extracellular diffusion, several studies highlighted that Dpp morphogen receptors (*Lecuit et al., 1996*; *Lecuit and Cohen, 1998*; *Lander et al., 2002*; *Crickmore and Mann, 2006*) and interaction partners (e.g. Dally, *Belenkaya et al., 2004*, *Akiyama et al., 2008*) found along the extracellular surface of target cells crucially influence morphogen gradient shape. Therefore, future studies will need to investigate the localization and the effect of forced mislocalization of Dpp receptors and interaction partners on Dpp dispersal and gradient formation. Furthermore, using a GrabFP toolset based on nanobodies or protein binders against other fluorescent proteins (*Brauchle et al., 2014*), the Dpp ligand and the Dpp receptors or interaction partners could be localized to different compartments and the effect of such altered localization could confirm or refute emerging hypotheses. Of course, it will be of critical importance to complement the results obtained using the GrabFP system with functional studies interfering with trafficking and secretion of Dpp.

## Material and methods

### Fly strains

The following fly lines were used: $y^1w^{1118}$ (wild-type control), Crb-GFP (*Huang et al., 2009*). *dpp:: LG86Fb* (K. Basler, *Yagi et al., 2010*), *LOP-eGFP-Dpp* and *LOP/UAS-morphotrap* (*Harmansa et al., 2015*), *tub>CD2,Stop>Gal4* (F. Pignioni), $sqh^{AX3}$ and *sqhSqh-GFP* (R. Karess) The fly stocks Dally-YFP, Dlp-YFP, Nrv1-YFP, Nrv2-YFP, NrxIV-YFP, Arm-YFP, αCat-YFP, Hts-YFP, Notch-YFP, Ed-YFP, PMCA-YFP have been obtained from the KYOTO Stock Center (DGRC) in Kyoto Institute of Technology. The fly line Fat-GFP is described in *Sarov et al. (2016)* and obtained from the VDRC stock center. *r4::Gal4* was obtained from Bloomington (BL33832). *nub::Gal4*, *ptc::Gal4*, *hh::Gal4*, $dpp^{d8}$ and $dpp^{d12}$ are described on FlyBase (www.flybase.org).

### Genotypes by figure

Figure 1: **C**, *nub::Gal4 / LOP/UAS-morphotrap*; **D**, *w; nub::Gal4 / LOP/UAS-GrabFP-A$_{Ext}$*; **E**, *w; nub:: Gal4 / LOP/UAS-GrabFP-B$_{Ext}$*;

Figure 2: **B**, *LOP/UAS-GrabFP-A$_{Ext}$ / +; NrxIV-YFP / hh::Gal4*; **C**, *LOP/UAS-GrabFP-A$_{Ext}$ / +; Dlp-YFP / hh::Gal4*; **D**, *LOP/UAS-GrabFP-B$_{Ext}$ / +; Crb-GFP / hh::Gal4*; **E**, *Notch-YFP / +; LOP/UAS-GrabFP-B$_{Ext}$ / +; hh::Gal4 / +*; **F**, *LOP/UAS-GrabFP-B$_{Ext}$ / Ed-YFP; hh::Gal4 / +*

Figure 3: **B**, *Fat-GFP / ptc::Gal4; LOP/UAS-GrabFP-B$_{Int}$ / +*; **C**, *ptc::Gal4 / +; LOP/UAS-GrabFP-B$_{Int}$ / αCat-YFP*; **D**, *Nrv1-YFP / ptc::Gal4; LOP/UAS-GrabFP-A$_{Int}$ / +*; **E**, *Nrv2-YFP / ptc::Gal4; LOP/UAS-GrabFP-A$_{Int}$ / +*; **F**, *Hts-YFP / ptc::Gal4; LOP/UAS-GrabFP-A$_{Int}$ / +*;

Figure 4: **A**, $sqh^{AX3}$ / +; *sqhSqh-GFP*; **B**, *ptc::Gal4 / +; LOP/UAS-GrabFP-B$_{Int}$ / +*; **D**, $sqh^{AX3}$ / +; *sqhSqh-GFP / ptc::Gal4; LOP/UAS-GrabFP-B$_{Int}$ / +*; **E–F**, $sqh^{AX3}$ / Y; *sqhSqh-GFP / ptc::Gal4; LOP/ UAS-GrabFP-B$_{Int}$ / +*

Figure 5: **A–B**: *w; LOP-eGFP-Dpp / +; dpp::LG86Fb / +*; **C–E**: *yw hsFlp; tub>CD2,Stop>Gal4, LOP-eGFP-Dpp / UAS-morphotrap; dpp::LG86Fb / +*

Figure 6: **A**: *w; LOP-eGFP-Dpp / +; dpp::LG86Fb / +*; **B**: *w; LOP-eGFP-Dpp / UAS-morphotrap; dpp::LG86Fb / hh::Gal4*; **C**: *w; LOP-eGFP-Dpp / UAS-GrabFP-B$_{Ext}$; dpp::LG86Fb / hh::Gal4*; **D**: *w; LOP-eGFP-Dpp / UAS-GrabFP-A$_{Ext}$; dpp::LG86Fb / hh::Gal4*;

Figure 7: **A,E,I,M**: *w; LOP-eGFP-Dpp, $dpp^{d12}$ / $dpp^{d8}$; dpp::LG86Fb / +*; **B,J,N**: *w; LOP-eGFP-Dpp, $dpp^{d12}$ / UAS*-morphotrap, $dpp^{d8}$; dpp::LG86Fb / hh::Gal4*; **F,K,O**: *w; LOP-eGFP-Dpp, $dpp^{d12}$ / UAS-GrabFP-B$_{Ext}$, $dpp^{d8}$; dpp::LG86Fb / hh::Gal4*;

Figure 8: **B**: *w; UAS-GrabFP-ECM / +; r4::Gal4 / +;* **D,F,I,M**: *w; LOP-eGFP-Dpp, dpp^{d12} / UAS-GrabFP-ECM, dpp^{d8}; r4::Gal4 / dpp::LG86Fb;* **E,H,L**: *w; LOP-eGFP-Dpp, dpp^{d12} / dpp^{d8}; dpp:: LG86Fb/ +*

## Molecular cloning
The following constructs were created using standard molecular cloning techniques.

### GrabFP-B_{Ext} - *pUASTLOTattB_vhhGFP4::Nrv1::TagBFP*
The TagBFP (Evrogen) coding sequence was inserted between the first and the second exon of the *nervana 1* (Nrv1, FlyBase ID: FBgn0015776) cDNA (BDGP DGC clone LD02379). The vhhGFP4 coding fragment (*Saerens et al., 2005*) was inserted at the C-terminal end of Nrv1::TagBFP. A *Drosophila* Kozak sequence (CAAA) was added and subsequently vhhGFP4::Nrv1::TagBFP was inserted into the multiple cloning site (MCS) of the pUASTLOTattB vector(*Kanca et al., 2014*).

### GrabFP-B_{Int} - *pUASTLOTattB_mCherry::Nrv1::vhhGFP4*
To generate a basolateral GrabFP construct that exposes the nanobody to the cytosol we started with the GrabFP-BExt plasmid. The tagBFP sequence was replaced by the vhhGFP4 sequence and the original vhhGFP4 sequence was exchanged with an mCherry coding sequence.

### GrabFP-A_{Ext} - *pUASTLOTattB_vhhGFP4::T48-Baz::mCherry*
The HA-tag was replaced by vhhGFP4 in the T48-HA plasmid (obtained from *Kölsch et al. [2007]*). mCherry was inserted at the C-terminal end of vhhGFP4::T48. In addition, the 2316 base pair minimal apical localization sequence of Bazooka (obtained from *Krahn et al., 2010*) was attached C-terminally to mCherry. A *Drosophila* Kozak sequence (CAAA) was added when inserting vhhGFP4::T48-Baz::mCherry into the MCS of the pUASTLOTattB vector (*Kanca et al., 2014*).

### GrabFP-A_{Int} - *pUASTLOTattB_mCherry::T48-Baz::vhhGFP4*
To switch the topology, we exchanged the mCherry with the vhhGFP4 coding region, resulting in orientation of the nanobody into the cytosol.

### GrabFP-ECM - *pUASTLOTattB_vhhGFP4::Vkg::mCherry*
vhhGFP4 and mCherry coding sequences, separated by a short linker region, were inserted between the first and second exon in the Vkg full-length plasmid (obtained from *Wang et al., 2008*). This insertion site was chosen, since a viable Vkg GFP-trap line exists which carries an exogenous GFP exon at this position (*Morin et al., 2001*). Finally the vhhGFP4::Vkg::mCherry construct was inserted into the MCS of the pUASTLOTattB vector (*Kanca et al., 2014*).

All transgenes were inserted by phiC31-integrase-mediated recombination into the 35B region on the second chromosome and the 86F region on the third chromosome (*Bischof et al., 2007*). The obtained transgenic flies respond to both, LexA and Gal4 transcriptional activators. By crossing with Cre^{y} expressing flies one of the response elements can be removed in a mutually exclusive manner. The excision was screened for by PCR as described in *Kanca et al. (2014)*.

## Antibodies
Rabbit (rb)-anti-mCherry (1:5000, gift from E. Nigg), rb-anti-tRFP (1:2000, Evrogen, #AB233), mouse (m)-anti-Dlg (4F3, 1:500, DSHB, University of Iowa), rb-anti-phospho-Smad1/5 (1:300; Cell Signaling, 9516S), guinea pig (gp)-anti Dll (1:2000, a gift from R. Mann), m-anti-Wg (4D4-s; 1:120; DSHB, University of Iowa); m-anti-Ptc (Apa1-s; 1:40; DSHB, University of Iowa). Secondary antibodies from the AlexaFluor series were used at 1:750 dilution with the exception from Alexa405-anti-rb which was used at 1:500 dilution. CF405S-anti-gp was used at 1:1000 dilution (Sigma Aldrich).

## Statistics and data representation
Sample size was chosen large enough ($n \geq 5$) to allow assessing statistical significance using a two-sided Student's *t*-test with unequal variance (*$p \leq 0.05$, **$p \leq 0.005$, ***$p \leq 0.0005$). Sample number and p-values are indicated in either the figure or the figure legend for each experiment. n-numbers indicate biological replicates, meaning the number of biological specimens evaluated (e.g. the

number of wing discs or wings). In boxplot graphs outliers are indicated by a red cross (e.g. *Figure 1—figure supplement 2*) and were excluded from further computation.

In the apical-basal concentration profiles (e.g. *Figure 2B–F*), bold lines represent average fluorescent values and error bars correspond to the standard error, dashed lines represent profiles corrected for the increase in GFP/YFP fluorescence levels due to nanobody binding (see below). Bold lines in the p-Mad expression profiles (*Figure 7D,H* and *Figure 8G*) indicate the arithmetic mean and the error bars show the standard deviation; individual profiles used for the analysis are shown light-colored. In box plots individual data points are shown and the center value represents the media while the whiskers mark the maximum and minimum data points.

## Specific methods for part I - fluorescent protein mislocalization (*Figures 2–4*)

### Sample collection, immunostaining and imaging

Third instar wandering larvae were dissected and used for analysis. Larvae were dissected, fixed and stained as described before (*Harmansa et al., 2015*). For high-resolution imaging along the z-axis (optical cross-sections of wing discs) discs were mounted with their apical side facing the coverslip and using double sided tape as spacer to avoid squeezing of the discs and to preserve their morphology. To obtain maximum resolution along the z-axis, stacks were acquired with sections every 0.17 μm on a Leica SP5 microscope. Importantly, due to our experimental design the GrabFP tools were only expressed in a subset of cells (the posterior compartment for GrabFP$_{Ext}$ and the *ptc*-domain for GrabFP$_{Int}$) and we used the non-GrabFP expressing cells as internal controls. Microscope settings were chosen to allow highest fluorescence levels (usually in the GrabFP domain) to be imaged under non-saturating conditions and were kept identical while imaging of all wing discs of one experiment (one target protein). Therefore, the fluorescence levels between the control and the experimental condition can be compared directly.

### Image processing

Image data was processed and quantified using ImageJ software (National Institute of Health). Optical cross-sections were computed using the section tool in Imaris software (Bitplane).

For improved resolution, datasets in *Figures 1–4* were deconvolved using the Huygens Remote Manager software (*Ponti et al., 2007*).

### Purification of eGFP and vhhGPF4 and fluorescent in vitro essay

In order to control for potential modulation of fluorescence signal due to GFP/YFP binding to vhhGPF4, we purified eGFP and vhhGFP4 and tested the effect of vhhGFP4 binding to eGFP in vitro. To do so, the coding sequences of vhhGFP4 and eGFP were cloned into pET22b(+) (Novagen) via NdeI and XhoI restriction enzyme sites. vhhGFP4 and eGFP proteins were expressed in BL21(DE3) *E. coli* bacteria (NEB) for 3 hr at 30°C using 1 mM IPTG. Subsequently, the respective bacterial pellets were lysed using a conventional Sonicator. The lysates were loaded on Protino Ni-NTA Agarose beads (Macherey-Nagel), and the proteins were purified according to the manufacturer's protocol. The purified proteins were dialyzed against 1x Phosphate Buffered Saline (PBS, Gibco) using Spectra/Por Dialysis Tubes (MWCO: 8000–10000 D, Spectrum Laboratories, Inc.).

To estimate changes in eGFP fluorescence upon vhhGFP4 binding we titrated defined amounts of purified vhhGFP4 (5–216 nM) on 54 nM purified eGFP in PBS (see *Figure 2—figure supplement 2*). Five independent replicas of the different concentration ratios were mixes in 96-well cell culture plates and incubated at room temperature for 30 min. eGFP fluorescence was imaged using a Leica SP5 confocal microscope. Fluorescent levels were measured using the histogram function (Analyze>Histogram) in ImageJ software (National Institute of Health) and average fluorescence values were plotted in Matlab software (Matworks). Our results show that low levels of vhhGFP4 (5–27 nM) do not result in a change of eGFP fluorescence (*Figure 2—figure supplement 2C*). However, vhhGFP4 concentrations of 54 nM or higher resulted in increased eGFP fluorescence. This increase plateaued for amounts of 108 nM and above (corresponding to a 1:2 ratio of eGFP to vhhGFP4). To estimate the mean increase in eGFP fluorescence when the plateau was reached (at saturating conditions), we calculated the mean value of all data points at plateau conditions (108 and 216 nM vhhGFP4); which resulted in a mean eGFP fluorescence increase of 47.5% or 1.475-fold under saturating conditions.

In order to give a fair representation of this phenomenon, we also show fluorescence profiles corrected for this 47.5% increase in fluorescence due to vhhGFP4 binding, as explained in the next section.

Furthermore, to ensure that we imaged eGFP fluorescence under microscope settings that are within the linear range of the fluorophores, we imaged defined amounts of purified eGFP protein under the identical setup as used for the in vitro fluorescence assay (*Figure 2—figure supplement 2E*). Indeed, the obtained fluorescent levels behaved proportional to the actual eGFP concentration, suggesting that our imaging conditions are in the linear-range and that eGFP fluorescence is proportional to eGFP concentration.

## Extraction of concentration profiles along the apical-basal axis

In order to quantify absolute protein levels in the apical versus the basolateral compartment, we acquired high-z-resolution stacks (as described in the imaging section) of wing discs stained for the junctional marked discs-large (Dlg). From these discs, we obtained optical cross-sections in the dorsal compartment parallel to the D/V boundary using the 'reslice' option in Fiji software (ImageJ, National Institute of Health) (see *Figure 1—figure supplement 3A*). From these cross-sections, we extracted the fluorescent intensity profiles of Dlg and the protein of interest in a rectangular region of 114 × 16 µm using the 'plot profile' function in ImageJ (see *Figure 1—figure supplement 3B*). Importantly, we extracted concentration profiles of the GrabFP expressing regions (experiment) and the non-GrabFP expressing neighboring cells (internal control) of the same discs. In order to average individual profiles from different discs, we used the junctional peak of the Dlg profile to align the individual profiles. To correct for variation between profiles from different discs, we subtracted the background fluorescence observed in the disc lumen (minimal fluorescence intensity observed in the luminal region, see *Figure 1—figure supplement 3A*). Average profiles were calculated in Excel software (Microsoft) and plotted in Matlab software (Matworks). In the depicted plots, we only included signal from the DP region and excluded signal from the PPE (see *Figure 1—figure supplement 3C–D*). The peak of the average Dlg profile plus and minus 1.0 µm (marked by a blue bar) was defined as the junctional plane and the border between the apical and the basolateral compartment. Error bars show the standard error.

In order to account for the observed increase in eGFP fluorescence due to interaction with vhhGFP4 in vitro, we show the original, non-corrected profiles (continuous lines) and profiles that were corrected for the observed increase in fluorescence due to vhhGFP4 (dashed lines). To do so, we corrected for the 1.475-fold increase incorporating the local vhhGFP4 (GrabFP) concentration using the following formula:

$$eGFPc(x) = eGFP(x) - \frac{eGFP(x) * relGrabFP(x)}{1.475} \qquad (1)$$

where eGFPc(*x*) is the corrected eGFP fluorescence, eGFP(*x*) the observed eGFP fluorescence and relGrabFP(*x*) the relative GrabFP concentration at position *x*. relGrabFP was calculated by normalization to maximum GrabFP fluorescence levels. Doing so, we correct eGFP fluorescence proportional to vhhGFP4 concentrations and only at positions where vhhGFP4 is present and potentially modifies eGFP fluorescence. Importantly, corrected profiles are supposed to provide means to account for a potential modulation of eGFP fluorescence by vhhGFP4 binding in vivo, based on our in vitro findings.

## Quantification of apical cell surface area

In order to assess the size change in apical cell surface area induced by Sqh-GFP basolateral mislocalization, we measured the apical area using the polygon selection tool in using ImageJ software (National Institute of Health). Individual data points were plotted in Matlab software (Matworks) using the Scatplot script (A. Sanchez-Barba; http://www.mathworks.com/matlabcentral/fileexchange/8577-scatplot).

## Specific methods for part II – Dpp gradient formation (*Figures 5–8* )

### Staging of larvae and dataset creation

For quantification of expression profiles (*Figure 7A–H* and *Figure 8E–G*) or pouch size (*Figure 7I–L* and *Figure 8H–J*) larvae were staged to 98–100 hr after egg laying (AEL) as described before (*Hamaratoglu et al., 2011*; *Harmansa et al., 2015*). Only male larvae were included in this analysis, positively selected by the presence of the transparent genitalia disc. All larvae of one experiment (control condition and one or several GrabFP conditions) were dissected and stained together using identical solutions, as described before (*Harmansa et al., 2015*). All wing discs of one experiment were mounted on the same cover slide using larval brains as spacers. Disc were mounted with the apical side of the DP facing the coverslip.

Data-sets were imaged in a SP5 confocal microscope (imaging a slice every 1 μm). All images of one data-set were acquired in the same microscope session using identical microscope settings; conditions were chosen within the linear range of the fluorescent signal obtained.

### Plotting of average expression profiles along the A-P axis

Average expression profiles were obtained using the WingJ software (*Schaffter, 2014*) (http://tschaffter.ch/projects/wingj/) as done previously (*Harmansa et al., 2015*). In brief, we made use of Wg and Ptc stainings marking the D/V and the A/P boundary, respectively. Profiles were then extracted up to the edge of the wing disc with a 30% offset in the dorsal compartment along a line parallel to the D/V boundary (see *Hamaratoglu et al., 2011*; *Harmansa et al., 2015*). Plotting of the average profiles was done in Matlab software (Mathworks) using the WingJ Matlab toolbox.

### Quantification of pouch and wing area

Areas of the wing pouch and the adult wing were quantified using the polygon selection tool in ImageJ software.

## Acknowledgements

We thank F Hamaratoglu, K Basler and G Pyrowolakis for discussions and input on the project, the Biozentrum Imaging Core Facility for maintenance of microscopes and technical support. We are grateful to M Leptin, A Wodarz, L Ashe, E Nigg, Y Hong and R Mann for flies and reagents.

## Additional information

### Funding

| Funder | Grant reference number | Author |
|---|---|---|
| SystemsX.ch Initiative (MorphogenetiX) | | Markus Affolter |
| Schweizerischer Nationalfonds zur Förderung der Wissenschaftlichen Forschung | | Markus Affolter |
| Grants from cantons Basel-Stadt and Basel-Land | | Markus Affolter |
| University of Basel | 'Fellowships for Excellence' PhD Program of the Biozentrum | Stefan Harmansa Ilaria Alborelli |

The funders had no role in study design, data collection and interpretation, or the decision to submit the work for publication.

### Author contributions

SH, IA, Conceptualization, Methodology, Investigation, Validation, Formal analysis and visualization, Writing—original draft, Writing—review and editing; DB, Methodology, Writing—review and editing; EC, Conceptualization, Supervision, Writing—review and editing; MA, Conceptualization, Supervision, Funding acquisition, Writing—review and editing

## Author ORCIDs

Stefan Harmansa, http://orcid.org/0000-0001-6668-7608
Ilaria Alborelli, http://orcid.org/0000-0001-7849-3958
Markus Affolter, http://orcid.org/0000-0002-5171-0016

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
