## [Decision Letter]

Thank you for submitting your article "A nanobody-based toolset to investigate the role of protein localization and dispersal in *Drosophila*" for consideration by *eLife*. Your article has been reviewed by two peer reviewers, and the evaluation has been overseen by K VijayRaghavan as the Reviewing Editor and Senior Editor. The following individual involved in review of your submission has agreed to reveal his identity: Frank Schnorrer (Reviewer #1).

The reviewers have discussed the reviews with one another and the Reviewing Editor has drafted this decision to help you prepare a revised submission.

Summary:

In this manuscript Affolter and colleagues expand their previous studies on nanobody-based protein trapping in *Drosophila*. They produce 5 novel synthetic GFP traps, together named 'Grab-FP toolbox' that trap GFP-fusion proteins everywhere at the cell membrane or preferentially at the apical or basolateral side of polarised epithelial cells. For each, 2 versions were developed with the trap facing intracellularly or extracellularly.

These new tools were used to re-localise GFP fusion proteins expressed in the wing disc epitelium at endogenous levels. For example, 8 of 8 transmembrane GFP fusion proteins were re-localised using the extracellular Grab-FP. This was particularly effective when re-localising from the apical to the basolateral membrane.

Also, cytoplasmic proteins can be re-localised. An impressive example was Myosin light chain, Sqh-GFP, that led to basolateral constriction of the cell when re-localised basally. This shows the power of the technique used here.

Finally, they used the basolateral extracellular trap to show that Dpp-GFP, expressed with dpp-lexA largely travels basolaterally to produce the Dpp protein gradient.

In summary, the experiments are well designed and the data support the authors' conclusions. I am impressed by the high quality of the data shown, in particular the quantifications of the immuno-fluorescent images

Essential revisions:

Concern 1.

Unfortunately, the functional consequences of protein mis-localisation were only investigated for Sqh-GFP and Dpp-GFP. As most of the Dpp is found basolaterally, it is not surprising that trapping it basolaterally produces the same phenotype as trapping it everywhere at the membrane (which was beautifully shown before), so the novelty is limited. As the authors discuss, specific trapping at the apical membrane would have been interesting, but was not possible with their system.

In order to make a strong point of the general usefulness of these new tools, it would be interesting to report if mis-localisation of the transmembrane proteins tested here like Crumbs, Notch etc. result in any functional consequences. Data from Berry et al. 2016 *eLife* argue that many proteins are surprisingly tolerant for moving them elsewhere in the cell. What do the authors see?

Concern 2.

2A. Other issues concern the quantification of the G/YFP signal and concern mainly Figure 2 and Figure 3, and their supplementary figures (micrographs and plots showing relative GFP concentration/signal). Can the authors show (or cite work that shows) that binding of the nanobody does not affect GFP/YFP signal intensity?

2B. The authors show rel. G/YFP signal in Figure 2, Figure 3 and supplementary figures. Relative to what is not clear. In most (but not all) cases it appears that the highest value along the a-b/l axis was set to 1. Absolute values measured in the same parallel experiments would seem more informative, in particular because it is not known whether the changes in distribution are caused by relocalization to another compartment or by stabilization of the fraction in the target compartment.

The Methods section states that images were acquired in the same session with the same solutions (for the 1um sections). However, it seems that confocal microscope settings were not identical and that they were optimized for individual pictures ("imaging conditions within the linear range…"). This would prevent a quantitative comparison between the control and the experimental sample.

Also, it is unclear, which of the two immunostaining paragraphs described in the Methods section applies to Figure 2 and Figure 3.

2C. Given the problems comparing experimental and control quantifications, Figures G and H are not adequate. They would be appropriate for absolute values acquired under identical conditions (but not if individual pictures were adjusted to using the entire dynamic range). For these reasons, the 2.8 and 12.6-fold enrichment, mentioned in the third paragraph of the subsection “Mislocalizing transmembrane and cytosolic proteins along the A-B axis using the GrabFP system” are overinterpretations. The same applies to the GrabFP-B value.

Micrographs and intensity curves in these figures are somewhat confusing because at first sight they often seem not to match (e.g. in many cases the GrabFP panels show different GFP intensities in the basolateral region than controls, but the red and black curves show the opposite difference for this region; see Figure 2, Figure 3).

The authors should be able to come up with an appropriate improvement on these points without necessarily doing additional experiments. One reviewer was not able to unambiguously figure out how they acquired their pics. Ideally, they have used the same settings in the experimental and corresponding control session and the settings were chosen such that even the strongest signal remained in the linear range. However, because they do not describe it in this way, it is possible that they either simply forgot to mention this one point or that they did not do it this way. In theory, you would not need an internal fluorescent standard with this approach if the disc-to-disc variations from the same experiment remained reasonable (because they used the same solutions and mounted the samples on the same slide). In short, if they have more "absolute" data that does not show too much variation, they should show it. Otherwise they should simply address these issues, trying to improve Figure 2 and Figure 3 where they can and briefly discuss the issues.

Concern 3.

Figure 5 / eGFP-Dpp studies.

In the second paragraph of the subsection Dpp disperses in the apical and in the basolateral compartment”, the authors claim that they did not detect eGFP-Dpp in the lumen. It is difficult to quantify luminal signals because they easily get washed out during fixation and staining unless they are crosslinked to a membrane. This is also suggested by the results in Figure 6. An additional control would be needed to make these claims.

---

## [Author Response]

*Essential revisions:*

*Concern 1.*

*Unfortunately, the functional consequences of protein mis-localisation were only investigated for Sqh-GFP and Dpp-GFP. As most of the Dpp is found basolaterally, it is not surprising that trapping it basolaterally produces the same phenotype as trapping it everywhere at the membrane (which was beautifully shown before), so the novelty is limited. As the authors discuss, specific trapping at the apical membrane would have been interesting, but was not possible with their system.*

We observe that Dpp is mainly found basolateral, and hence our finding that basolateral trapping results in a severe phenotype is not surprising (based on the observed Dpp localization). However, we think that this does not limit the novelty, since these findings address a longstanding question in the field, namely if the functional Dpp morphogen gradient forms in the apical/luminal or in the basolateral compartment (Entchev, Schwabedissen et al. 2000, Teleman and Cohen 2000, Gibson, Lehman et al. 2002). More importantly, this is the first time that we have functionally interfered with specific sub-pools of the Dpp morphogen and directly access their respective contribution to patterning and growth. We are convinced that the tools and concepts we describe in this manuscript provide the research community with powerful novel means to address similar questions regarding the spreading of other morphogens, such as Hedgehog and Wingless and other secreted proteins.

*In order to make a strong point of the general usefulness of these new tools, it would be interesting to report if mis-localisation of the transmembrane proteins tested here like Crumbs, Notch etc. result in any functional consequences. Data from Berry et al. 2016 eLife argue that many proteins are surprisingly tolerant for moving them elsewhere in the cell. What do the authors see?*

We now have performed experiments in which we mislocalize the apical proteins Crumbs-GFP or Notch-YFP in homozygous GFP/YFP-tagged backgrounds using Grab-B_Ext_. Indeed, we observe functional consequences due to protein mislocalization. Notch mislocalization has a strong phenotypical impact on wing disc shape. The pouch of Notch-YFP hemizygous males expressing GrabFP-B_Ext_ in their posterior compartment is severely stretched along the dorso-ventral axis (Figure 9). Surprisingly, we observe ectopic induction of the Notch target Wg along the anterior-posterior boundary (Figure 9). Ectopic expression of Wg along the A/P boundary results in up-regulation of the Wg target Distalless (Dll) (Figure RB1E mid panel). Furthermore, we observe increased cell division along the ectopic Wg stripe (Figure 9 right panel). This altered proliferation pattern might account for the elongated disc shape caused by N-YFP mislocalization. Interestingly, Wg is induced in a stripe of cells adjacent to the region expressing GrabFP-B_Ext_ (Figure 9). The molecular mechanism underlying Wg activation in this condition remains elusive and requires further in-depth investigations, which are out-of-the scope for the framework of this manuscript. However, it is known that Notch signalling is tightly regulated by the distribution of its ligand and interaction partners (Troost and Klein 2012 and several others) and that the apical localization of Notch receptor is required for Notch signalling activity (Sasaki, Sasamura et al. 2007). Accordingly, our data suggest that receptor function is influenced by the cellular environment and is drastically modified when pathway components are moved to a different cellular compartment.

Author response image 1.N-YFP mislocalization causes the induction of an ectopic Wg stripe.(**A**) Representative wing disc of a N-YFP (green) hemizygous male (control) stained for Wg (upper panel, blue; lower panel, black). Wg is expressed along the D/V boundary. (**B**) Basolateral mislocalization of N-YFP (green) in hemizygous male N-YFP/Y flies in the posterior compartment by GrabFP-B_Ext_ (red) leads to the induction of a secondary Wg stripe along the A/P boundary (upper panel, blu; lower panel, black). (**C**) Blow-up of region marked in (B), showing that Wg is only induced in the row of cells adjacent to the posterior compartment. D, Wing disc of a male control fly stained for the Notch target Distal-less (Dll, red) and the mitotic marker phosphor-Histone H3 (PH3, blue). The pouch outline is marked by a dashed line. Distal-less staining marks the oval pouch and the observed proliferation pattern is fairly uniform. E, Wing discs of two N-YFP hemizygous males expressing GrabFP-B_Ext_ in the posterior compartment. Ectopic Wg induction results in expansion of the pouch along the dorsal-ventral axis as shown by the expansion of the Dll staining. This change in pouch shape might be a reason of a non-uniform proliferation pattern in this background, with increased cell division along the ectopic Wg stripe.**DOI:**
http://dx.doi.org/10.7554/eLife.22549.023

We also tested the effect of Crb-GFP mislocalization on Crb interaction partners in a Crb-YFP homozygous background (PatJ, Ex, (reviewed in Bulgakova and Knust 2009 and Ling, Zheng et al. 2010) and other apical proteins (aPKC, Fat, see Hamaratoglu, Gajewski et al. 2009) that are known to require Crb for their proper localization. We observed a severe reduction in the apical localization of all protein tested, but no clear mislocalization to the basolateral side of the wing disc epithelia (Figure 10). In this specific case, it is proposed that Crb functions redundantly with other polarity determinants, such as Bazooka. Indeed, *crb* mutant epithelia (Pellikka, Tanentzapf et al. 2002, Tanentzapf and Tepass 2003) as well as cells with mislocalized Crb do not display obvious polarity defects.

Author response image 2.Mislocalization of Crb-GFP affects the localization of apical Crb interaction partners.(**A-D**) Optical cross-sections of Crb-GFP (green) homozygous GFP-tagged wing discs showing the anterior control compartment (Ctrl., left) and the posterior compartment expressing GrabFP-B_Ext_ (right). Discs were stained for the Crb interacting proteins PatJ (A), Expanded (Ex, B), Fat (C) and aPKC (D). In all cases, loss of apical Crbs due to mislocalization resulted in a striking reduction of the apical levels of all Crb interactors tested. Interestingly, no basolateral increase in PatJ, Ex, Fat or aPKC is observed, suggesting that there are control mechanisms excluding these proteins from the basolateral compartment.**DOI:**
http://dx.doi.org/10.7554/eLife.22549.024

As discussed in Berry, Olafsson et al. (and as the reviewers suggested), there are cases in which the cell is surprisingly tolerant to forced protein mislocalization. However, our data show that alteration in the subcellular protein distribution can result in dramatic phenotypic changes (see Sqh-GFP or Notch-YFP mislocalization). In conclusion, we believe that our first investigations show that protein mislocalization in a homozygous GFP/YFP-tagged background results in novel, not yet described phenotypes that clearly differ from classical genetic manipulation studies. This suggests that our tools allow addressing questions, which are difficult to answer using existing tools.

*Concern 2.*

*2A. Other issues concern the quantification of the G/YFP signal and concern mainly Figure 2 and Figure 3, and their supplementary figures (micrographs and plots showing relative GFP concentration/signal). Can the authors show (or cite work that shows) that binding of the nanobody does not affect GFP/YFP signal intensity?*

This is a very valid concern, that we have neglected in the first version of our manuscript. Indeed, it was shown before that nanobody binding can result in the modulation of GFP fluorescence properties (Kirchhofer, Helma et al. 2010). In order to address if binding of GFP to the nanobody we used (vhhGFP4) modifies the fluorescent properties of the former, we have now included the results of an in vitro fluorescence assay (Figure 2—figure supplement 2).

To estimate the changes in eGFP fluorescence upon binding of the nanobody, we have titrated purified nanobody on purified eGFP in vitro and measured the changes in fluorescence levels (see Figure 2—figure supplement 2, explained in detail in the Methods section: “Purification of eGFP and vhhGFP4 and fluorescence assay”). The results we obtain from five independent repetitions consistently show that binding of the nanobody to eGFP results in an average increase in fluorescence levels of roughly 50% under saturating conditions.

Importantly, the implications of this result on the conclusion of our experiments is limited to those observations in which existing fractions were modified in their levels. (such as seen for Dally or Dlp). In contrast, in most conditions the expression of the GrabFP system resulted in the gain of a novel fraction (as for NrxIV, Nrv1 and Nrv2) or the loss of an existing one (as for Crb, Notch, Ed or Fat). In the latter case, when a fraction is lost, increasing EGFP fluorescence due to nanobody binding does not affect the obtained interpretations. However, when gaining a novel fraction, we have to put into perspective that this novel fraction is overestimated by roughly 50% bigger due to the effects of nanobody-binding.

In order to provide a fair representation of this effect, we now included profiles which were corrected for the effect of nanobody binding on eGFP fluorescence (for detail see Methods section: “Extraction of concentration profiles along the apical-basal axis”). Therefore, we now show in our graphs both the original, non-corrected profiles (continuous lines) and the corrected profiles (dashed lines). In addition, all our statistical analysis is done on the corrected data to avoid any over interpretation of our data.

These findings are included in the main text:

“However, it is known that the binding of nanobodies can interfere with the fluorescent properties of GFP (Kirchhofer, Helma et al. 2010). […] To account for this likeliness in our quantifications, we included A-B profiles of the observed GFP/YFP fluorescence levels (continuous red line) as well as profiles that were corrected for a potential fluorescence increase at GrabFP-positive positions (dashed red line, see methods for details).”

*2B. The authors show rel. G/YFP signal in Figure 2, Figure 3 and supplementary figures. Relative to what is not clear. In most (but not all) cases it appears that the highest value along the a-b/l axis was set to 1. Absolute values measured in the same parallel experiments would seem more informative, in particular because it is not known whether the changes in distribution are caused by relocalization to another compartment or by stabilization of the fraction in the target compartment.*

*The Methods section states that images were acquired in the same session with the same solutions (for the 1um sections). However, it seems that confocal microscope settings were not identical and that they were optimized for individual pictures ("imaging conditions within the linear range…"). This would prevent a quantitative comparison between the control and the experimental sample.*

*Also, it is unclear, which of the two immunostaining paragraphs described in the Methods section applies to Figure 2 and Figure 3.*

We are sorry for this lack of clarity in the manuscript and the corresponding Methods section. To improve this, we have now split the methods in two parts, describing the procedures for the mislocalization part and the procedures for the Dpp part separately.

Indeed, all experiments for protein mislocalization were imaged under identical microscope setups. More precisely, since we only modified parts of the discs (either the Hh or the Ptc domain), we used the surrounding tissue as an internal control. Hence, fluorescence profiles were extracted from the same disc for the experiment and the control condition. Subsequent, we averaged data obtained from several discs (all processed in the same tube and imaged under identical conditions and in the same session) to obtain the plots and quantifications shown in Figure 2 and Figure 3 and the corresponding supplementary figures. This procedure is now mentioned in the main text and explained in detail in the Methods section:

Subsection “Mislocalizing transmembrane and cytosolic proteins along the A-B axis using the GrabFP system”:

“Therefore, single components of the GrabFP system were co-expressed with different target proteins in defined domains of the wing imaginal disc (*hh::Gal4* for GrabFP_Ext_ and *ptc::Gal4* for GrabFP_Int_), while neighbouring areas were used as an internal control for the analysis of wild-type target protein localization.”

Subsection “Sample collection, immunostaining and imaging”:

“Importantly, due to our experimental design the GrabFP tools were only expressed in a subset of cells (the posterior compartment for GrabFP_Ext_ and the *ptc*-domain for GrabFP_Int_) and we used the non-GrabFP expressing cells as internal controls. […] Therefore, the fluorescence levels between the control and the experimental condition can be compared directly.”

And subsection “Extraction of concentration profiles along the apical-basal axis”:

“Importantly, we extracted concentration profiles of the GrabFP expressing regions (experiment) and the non-GrabFP expressing neighbouring cells (internal control) of the same discs.”

*2C. Given the problems comparing experimental and control quantifications, Figures G and H are not adequate. They would be appropriate for absolute values acquired under identical conditions (but not if individual pictures were adjusted to using the entire dynamic range). For these reasons, the 2.8 and 12.6-fold enrichment, mentioned in the third paragraph of the subsection “Mislocalizing transmembrane and cytosolic proteins along the A-B axis using the GrabFP system”, are overinterpretations. The same applies to the GrabFP-B value.*

*Micrographs and intensity curves in these figures are somewhat confusing because at first sight they often seem not to match (e.g. in many cases the GrabFP panels show different GFP intensities in the basolateral region than controls, but the red and black curves show the opposite difference for this region; see Figure 2, Figure 3).*

As stated above, GFP signals for control and experimental conditions were extracted from discs that were processed and imaged under identical conditions. Hence, a direct comparison of fluorescence levels is possible and fair. We originally chose this kind of representation solely to visualize the relative changes in fluorescence levels between the basolateral and the apical compartment. Therefore, the black and the white curve were not related in their levels. We realized that this representation might be confusing to the reader as mentioned by the referee.

To improve this point, we now included plots showing absolute fluorescence levels (instead of relative, as suggested by the referees). Now levels between control and experimental graphs can be directly compared and statistical significant changes are directly indicated in the plots. We therefore also omitted to show a separate quantification of fluorescent levels (former bar graphs in Figure 2 and Figure 3).

*The authors should be able to come up with an appropriate improvement on these points without necessarily doing additional experiments. One reviewer was not able to unambiguously figure out how they acquired their pics. Ideally, they have used the same settings in the experimental and corresponding control session and the settings were chosen such that even the strongest signal remained in the linear range. However, because they do not describe it in this way, it is possible that they either simply forgot to mention this one point or that they did not do it this way. In theory, you would not need an internal fluorescent standard with this approach if the disc-to-disc variations from the same experiment remained reasonable (because they used the same solutions and mounted the samples on the same slide). In short, if they have more "absolute" data that does not show too much variation, they should show it. Otherwise they should simply address these issues, trying to improve Figure 2 and Figure 3 where they can and briefly discuss the issues.*

In summary, we have performed an additional in vitro essay to test for a potential effect of nanobody binding on eGFP fluorescence. Indeed, we find that binding of vhhGFP4 to eGFP results in an average increase of 47.5% in eGFP fluorescence in vitro. We have accounted for this effect by additionally showing profiles corrected for this increase. Furthermore, we now included profiles showing absolute eGFP/YFP levels that allow direct comparison of concentrations along the apical-basal axis. We are convinced that these changes improved the representation and clarity of our data.

Concern 3.

Figure 5 / eGFP-Dpp studies.

*In the second paragraph of the subsection Dpp disperses in the apical and in the basolateral compartment”, the authors claim that they did not detect eGFP-Dpp in the lumen. It is difficult to quantify luminal signals because they easily get washed out during fixation and staining unless they are crosslinked to a membrane. This is also suggested by the results in Figure 6. An additional control would be needed to make these claims.*

We stated that: “We did not detect eGFP-Dpp signal within the luminal space (Figure 5, magnification). These results suggest that, using fluorescence microscopy, Dpp is prominently detected within the lateral plane of the DP epithelium.” Importantly, we here state that under our experimental setup and detection method we were not able to detect apical/luminal eGFP::Dpp; we did not want to imply that there is none. The comment of the referee is very important and it seems that luminal eGFP-Dpp, if present in the luminal cavity at significant levels, is extremely difficult to detect. We also tried to use extracellular staining protocols to visualize luminal Dpp. In short, live discs are incubated in antibody solution and subsequently excessive antibody is washed of and discs are fixed. However, also using this approach we did not detect apical/luminal Dpp, possibly because the antibody cannot penetrate the lumen without permeabilization (results not shown). Hence, it seems that the lumen is not accessible for antibodies without permeabilization, however, once permeabilized, non surface-bound eGFP-Dpp is possible washed off the disc, as suggested by the reviewer.

To control for this possibility, we expressed a secreted vhhGFP::mCherry fusion protein (secVHH-mCherry) in the posterior cells of the wing disc (Figure 11). secVHH-mCherry is secreted into the disc lumen and disperses to ultimately fill the whole luminal cavity. Importantly, secVHH-mCherry can be easily detected by a conventional immunostaining approach and is not washed off despite cell permeabilization. These results suggest that a small protein fused to a fluorophore (secVHH-mCherry is of similar size as eGFP::Dpp) is retained in the wing disc lumen and can be detected by a standard immunostaining protocol. Most importantly, once we used nanobody-mediated immobilization in clones or in the peripodial epithelium, low levels of apical/luminal eGFP-Dpp can be visualized. This suggests that apical luminal levels are too low to be detected without the stabilizing effect of the GrabFP system. Such effect should be further amplified due to the increase in eGFP fluorescence caused by the binding of the nanobody.

Author response image 3.Detection of a secreted VHH-mCherry fusion in the wing disc lumen.(**A**) Schematic representation of the experimental setup. We expressed a fusion of the mouse CD8 signal peptide, the vhhGFP4 GFP-nanobody and mCherry (secretedVHH-mCherry) under the control of *hh::Gal4* in the posterior compartment of otherwise wild type wing discs. Importantly, the anterior/posterior compartment boundary (A/P) in the peripodial epithelium (top layer) is shifted anteriorly compared to the disc proper (bottom layer). However, *hh::Gal4* is exclusively active in posterior cells. (**B**) Optical cross-sections of wing discs expressing secretedVHH-mCherry (green) in the posterior compartment (*hh::Gal4*). The wing disc shown was permeabilized by several washes with PBST and subsequently stained for DAPI (magenta). Despite the permeabilization, secretedVHH-mCherry was not washed out and can even be detected in the anterior most parts of the lumen (arrow).**DOI:**
http://dx.doi.org/10.7554/eLife.22549.025